# A plant Bro1 domain protein BRAF regulates multivesicular body biogenesis and membrane protein homeostasis

Jinbo Shen [1,5], Qiong Zhao [1], Xiangfeng Wang [1,2], Caiji Gao[1,3], Ying Zhu[1], Yonglun Zeng[1] & Liwen Jiang[1,4]

Plant development, defense, and many physiological processes rely on the endosomal sorting complex required for transport (ESCRT) machinery to control the homeostasis of membrane proteins by selective vacuolar degradation. Although ESCRT core components are conserved among higher eukaryotes, the regulators that control the function of the ESCRT machinery remain elusive. We recently identified a plant-specific ESCRT component, FREE1, that is essential for multivesicular body/prevacuolar compartment (MVB/PVC) biogenesis and vacuolar sorting of membrane proteins. Here we identify a plant-specific Bro1-domain protein BRAF, which regulates FREE1 recruitment to the MVB/PVC membrane by competitively binding to the ESCRT-I component Vps23. Altogether, we have successfully identified a role for BRAF, whose function as a unique evolutionary ESCRT regulator in orchestrating intraluminal vesicle formation in MVB/PVCs and the sorting of membrane proteins for degradation in plants makes it an important regulatory mechanism underlying the ESCRT machinery in higher eukaryotes.

[1] Centre for Cell & Developmental Biology, State Key Laboratory of Agrobiotechnology, School of Life Sciences, The Chinese University of Hong Kong, Shatin, New Territories, Hong Kong, China. [2] State Key Laboratory of Plant Physiology and Biochemistry, Department of Plant Sciences, College of Biological Sciences, China Agricultural University, Beijing 100193, China. [3] Guangdong Provincial Key Laboratory of Biotechnology for Plant Development, School of Life Sciences, South China Normal University (SCNU), Guangzhou 510631, China. [4] CUHK Shenzhen Research Institute, The Chinese University of Hong Kong, Shenzhen 518057, China. [5] Present address: State Key Laboratory of Subtropical Silviculture, Zhejiang A&F University, Linan, Hangzhou 311300, China. These authors contributed equally: Jinbo Shen, Qiong Zhao. Correspondence and requests for materials should be addressed to J.S. (email: jshen@zafu.edu.cn) or to L.J. (email: ljiang@cuhk.edu.hk)

The abundance and localization of integral plasma membrane (PM) proteins, including signaling receptors, ion channels, and nutrient transporters, allow for multiple physiological functions in growth, differentiation, and survival of eukaryotic cells. Thus, the tight control of membrane protein homeostasis by selective vacuole/lysosome degradation is not only essential for the destruction of non-functional or misfolded proteins but also ensures proper cell signaling and facilitates interactions with the environment[1,2]. In this sorting process, membrane proteins are firstly ubiquitinated and subsequently sequestered into the intraluminal vesicles (ILVs) of the multivesicular bodies/prevacuolar compartments (MVB/PVCs) through the function of the endosomal sorting complex required for transport (ESCRT) machinery. Ultimately, the fusion of the MVB/PVCs allows the membrane proteins to be degraded in the lumen of the vacuole/lysosome[3,4].

The formation and scission of ILVs in MVB/PVC are mediated by the ESCRT machinery, which are assembled on the endosomal membrane into several protein complexes, termed ESCRT-0, -I, -II, -III, and the Vps4 complex[3,5–7]. The *Arabidopsis* genome contains most canonical ESCRT components, except for ESCRT-0 subunits and the ESCRT-I component Mvb12[8–11]. It is suggested that other components might therefore act as non-canonical ESCRT-0 in cargo recognition in plants, together with the unique plant Fab1, YOTB, Vac1, and EEA1 (FYVE) domain protein required for endosomal sorting 1 (FREE1 or FYVE1)[12,13]. FREE1 is a phosphatidylinositol 3-P-binding protein, which also interacts with the ESCRT-I complex component Vps23. Consistent with the ESCRT mutants phenotype, in which the assembly or dissociation of the ESCRT machinery is disrupted, FREE1 loss-of-function mutants (T-DNA and RNAi mutants) are seedling lethal, resulted from defects in the formation of ILVs in MVBs, which eventually block endocytosed PM proteins, such as the auxin efflux carrier PIN2 and iron transporter IRT1, gaining access to the vacuole lumen for degradation[12,13]. Although FREE1 has been assumed to be reserved for integral membrane proteins from the PM to vacuole degradation pathway, recent studies also show the involvement of FREE1 in the vacuolar degradation of the membrane associated ABA receptor, PYL4[14]. Moreover, it has also been shown that FREE1 manipulates autophagic degradation in plants by interacting with a unique plant autophagic regulator SH3 domain-containing protein 2 (SH3P2)[15,16]. Because of the multiple functions of FREE1 in *Arabidopsis*, it is likely that plants have evolved unique mechanisms and components regulating FREE1 functions.

In this study, by means of a screen for mutations that suppress the lethal phenotype of *FREE1-RNAi*, we identify the *sof524* mutant, which is due to a loss of function in BRAF, an *Arabidopsis* Bro1-domain protein. Through a combination of cellular, biochemical, and genetic approaches, we further demonstrate that BRAF and FREE1 compete for binding to ESCRT-I component Vps23 on MVB/PVCs and thus function as an important regulator for FREE1 function in ILV formation of MVB/PVC and membrane protein vacuolar sorting.

## Results

**sof524 can rescue FREE1-RNAi lethal plants.** With the aim of elucidating the molecular basis of FREE1 regulation, we have developed a genetic screen to search for *Arabidopsis sof* (suppressors of *free1*) mutants that suppressed the seedling lethal phenotype of FREE1 loss-of-function mutants (Supplementary Fig. 1A)[17]. Instead of using a *free1* T-DNA insertion mutant, which is hard to apply for suppressor screening of lethal mutant plants, we have taken advantage of dexamethasone (DEX)-inducible *DEX::FREE1-RNAi* lethality plants to collect true *sof*

mutants for gene identification. A *sof524* mutant was selected from the ethyl methanesulfonate (EMS)-mutagenized M2 population on the basis of a seedling survival phenotype upon DEX treatment. When *sof524* was grown on Murashige and Skoog (MS) medium or soil, there was no obvious phenotypic difference to the wild type (WT). However, after 7 days of growth on DEX medium, the *sof524* seedlings showed a recovered WT phenotype in contrast to the lethal phenotype of *FREE1-RNAi* seedlings, although the root length of *sof524* seedlings was reduced compared to the WT (Fig. 1a, b and Supplementary Fig. 1B). This result indicated that the *sof524* mutation partially reverts the deficient phenotype of *FREE1-RNAi*. Furthermore, the successful induction of the *FREE1* silencing system in *FREE1-RNAi* and *sof524* lines was confirmed by immunoblot analysis on seedlings on DEX-containing growth medium (Fig. 1c and Supplementary Fig. 1C). Taken together, these results suggested that the reverse phenotype of *sof524* is not caused by a disruption of the RNAi process.

Ubiquitinated membrane proteins are sorted into the ILVs of MVB/PVCs for further degradation upon MVB/PVC-vacuole fusion. According to our previously proposed model of FREE1 functions in the ILV formation and ubiquitinated membrane cargo sorting (Supplementary Fig. 2A)[12,13], depletion of FREE1 results in failure of ILV formation and consequently causes accumulation of ubiquitinated membrane cargo in endosomes and finally at the tonoplast. We then performed a series of experiments to test whether *sof524* mutants were able to restore the phenotype of *FREE1-RNAi* plants in membrane protein homeostasis. Consistent with controls of *FREE1-RNAi* without DEX induction, the DEX-treated *sof524* mutants expressing the auxin efflux carrier component PIN2-GFP were incubated in darkness for 6 h to stabilize the fluorescent proteins in the lytic vacuole[18], green fluorescent protein (GFP) signals were observed in the root vacuole with the endocytic tracker FM4-64 at the vacuolar membrane (Fig. 1d), suggesting that these proteins were indeed transported to the vacuole for degradation. In contrast, no obvious vacuolar accumulation of PIN2-GFP was observed in DEX-treated *FREE1-RNAi* plants, instead the internalized PIN2-GFP signal was found at the vacuolar membrane. In addition, similar results were observed for the recovery phenotype in boron-induced vacuolar degradation of the borate transporter BOR1-GFP in the *sof524* mutant (Fig. 1e). To further establish how the *sof524* mutant restores the defect of *FREE1-RNAi* plants in sorting ubiquitinated membrane cargo for vacuolar degradation, we conducted immunoblot assays with the ubiquitin antibody (anti-UBQ) on membrane protein extracts of *FREE1-RNAi* and *sof524* seedlings. DEX-treated *FREE1-RNAi* seedlings accumulated ubiquitinated proteins at a higher level than those without DEX treatment. Interestingly, accumulation of ubiquitinated proteins is restored to the WT level in *sof524* mutant plants (Fig. 1f). Taken together, these results suggest that the *sof524* mutant restored the defect(s) in global vacuolar degradation of ubiquitinated membrane proteins.

Successful vacuolar transport of ubiquitinated membrane proteins depends on the structure integrity of MVB/PVCs. We speculated that the *sof524* mutant after DEX treatment can still form typical MVB structures with numerous ILVs, and therefore investigated MVB morphology at the ultrastructural level by performing transmission electron microscopic (TEM) analysis of ultrathin sections of high-pressure frozen/freeze-substituted samples. Consistent with previous reports on the morphology of MVBs in *free1* mutant root cells, empty MVBs lacking ILVs, which are nicely labeled by the MVB/PVC marker vacuolar sorting receptor (VSR) antibodies, could be observed in *FREE1* RNAi cells (Fig. 1g). In contrast, VSR antibodies labeled typical MVB structures with numerous ILVs in the root cells of *sof524*

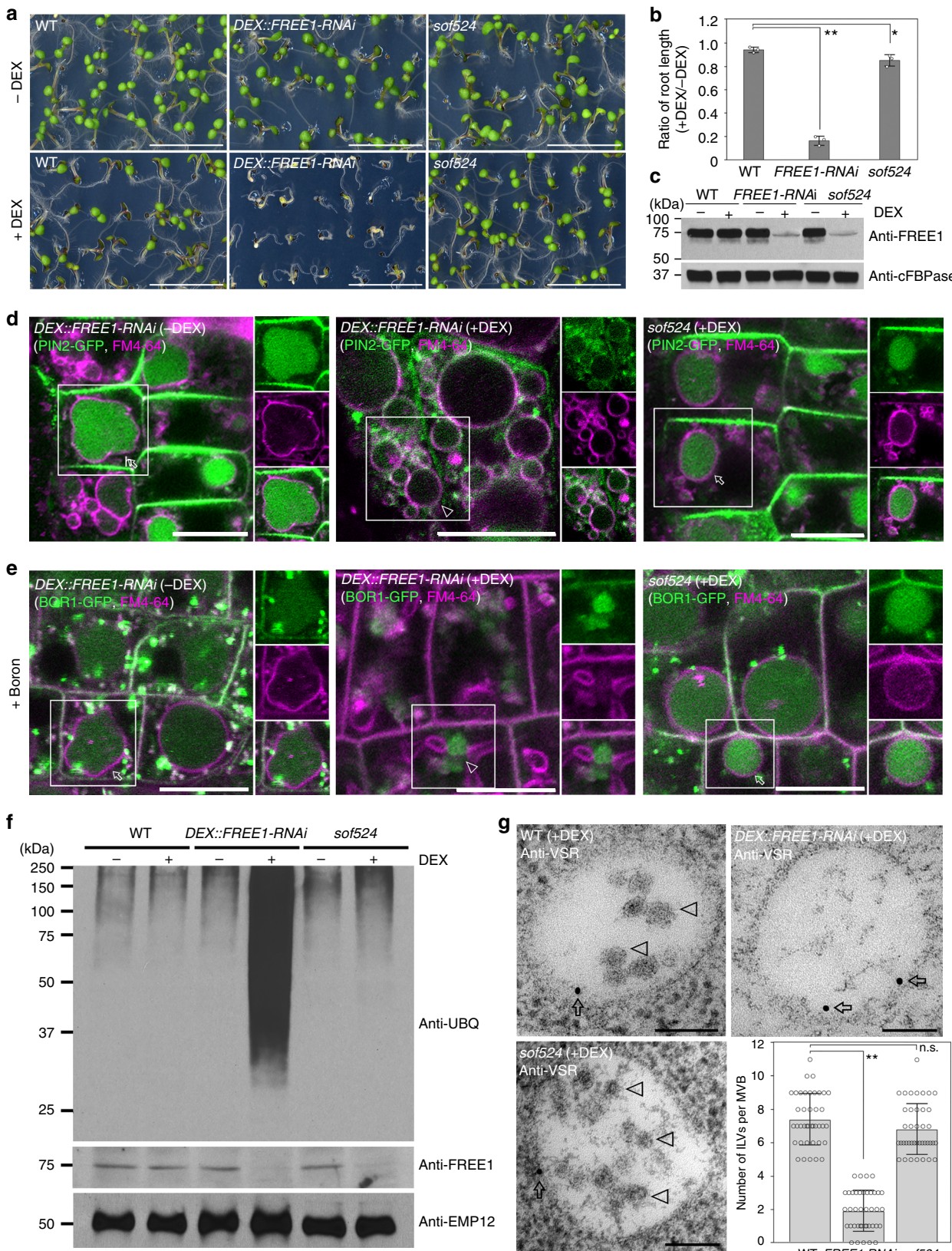

under DEX. A statistical analysis of the VSR antibody-labeled MVBs/PVCs further supports our speculation that the *sof524* mutant restored the integrity of ILVs in MVBs (Fig. 1g), which finally recovered ubiquitinated membrane cargo for vacuolar degradation. Moreover, cell biological analysis further revealed

that the *sof524* mutant partially restored the loss of the FREE1 phenotype in terms of central vacuole morphology as marked by the tonoplast marker YFP-VAMP711 and the soluble lytic vacuolar cargo spL-RFP transport into central vacuole (Supplementary Fig. 2B, C).

**Fig. 1** The *sof524* mutant rescues seedling lethality of *FREE1-RNAi*. **a** *sof524* M3 seeds plated on MS plates supplied with DEX can rescue the DEX-inducible *FREE1-RNAi* lethal seedlings. Scale bar, 1 cm. **b** The ratio of root length growth on MS plates supplied with DEX relative to without DEX in different genotypes for 7-day-old seedlings. Error bars are the S.D. from three independent experiments. *$P < 0.05$, **$P < 0.01$ in Student's *t*-test. **c** Reduced FREE1 protein level was detected in *sof524* mutant. Immunoblot analysis of protein extracts from 7-day-old seedlings of the indicated genotypes. The cytoplasmic marker anti-cFBPase, a ubiquitously expressed cytosolic fructose-1,6-bisphosphatase, is used as a loading control. **d** The auxin efflux carrier PIN2-GFP vacuolar degradation is converted in *sof524* mutant. Seven-day-old seedlings of indicated genotypes with(+)/without(−) DEX induction were incubated with FM4-64, and visualized after 6 h dark treatment. Arrows and arrowheads indicate the vacuolar GFP signal and tonoplast-localized GFP signal, respectively. Separated images of each channel in the white outline area are shown on the right side (from top to bottom: GFP, RFP, and merged). Scale bar, 10 μm. **e** Vacuolar degradation of the boron transporter BOR1-GFP is converted in the *sof524* mutant. Seven-day-old seedlings grown in low boron medium with(+)/without(−) DEX induction were transferred to high (+B) boron liquid media for 6 h incubation in dark and with equal time of FM4-64 uptake. Arrow and arrowhead indicate the vacuolar GFP signal and endosome-localized GFP signal, respectively. Scale bar, 10 μm. **f** The *sof524* mutant does not accumulate ubiquitin conjugates. Immunoblots of membrane protein extracts from 7-day-old indicated genotypes with(+)/without(−) DEX induction using anti-UBQ, anti-FREE1, or anti-EMP12. **g** Electron micrographs of ILV formation in MVBs. Ultrathin sections were prepared from HPF/FS samples of indicated genotype plant roots with (+) DEX induction, followed by immunogold labeling using VSR antibodies. The number of ILVs per MVB was statistically analyzed on 40 MVBs recognized by VSR antibodies. Error bars are the S.D. **$P < 0.01$; n.s., $P > 0.05$ in Student's *t*-test. Arrows and arrowheads indicate the gold particles and ILVs, respectively. Scale bar, 100 nm

**The *sof524* mutation affects the Bro1-domain protein BRAF.**
To identify the mutated gene, *sof524* was firstly outcrossed to *Ler*. In the F2 mapping population, we collected the 7-day-old survival seedlings on MS plates supplemented with DEX and hygromycin, and the whole genome was sequenced following the previously described workflow[17]. The distribution of seedlings with the survival phenotype (*sof524* homozygous for *FREE1-RNAi*) in the mapping population (211 out of 1327 plants, an expected 3/16 of total seedlings was able to survive) is consistent with the number expected for a recessive mutation not linked to the insertion site of the *FREE1-RNAi*. The next-generation sequencing (NGS) mapping of the whole-genome-sequencing sample identified a peak specific for *sof524* located on the left arm of chromosome 5 (Supplementary Fig. 3A), and the fine mapping identified a G-to-A transition in the open reading frame of *AT5G14020*. A single-nucleotide change (G989A) was identified in the ninth exon of the locus that produced a mis-sense mutation (alanine[330]-to-valine) in the predicted protein sequence (Fig. 2a). The *AT5G14020* gene encodes a Bro1-domain-containing protein (Supplementary Fig. 3B), the biological function of which was still unknown. Here we use the name *BRo1-domain protein As FREE1 suppressor* (*BRAF*) for its related function.

In the *Arabidopsis* genome, there are five genes that encode a Bro1-domain: *AtBRO1/ALIX*, *BRAF*, *AT1G13310*, *AT1G17940*, and *AT1G73990*. AtBRO1/ALIX corresponds to a conserved multifunctional class of proteins extensively studied in yeast (represented by Bro1p and Rim20) and mammals (ALIX), which functions to recruit the de-ubiquitinating enzyme to remove ubiquitin from cargo[19–23] and also works for the selective sorting of ubiquitinated cargo at an early step in endocytosis[24,25]. Depletion of AtBRO1/ALIX in plants affects the seed-to-seedling developmental phase transition[22,23,25]. AtBRO1/ALIX has an N-terminal Bro1-domain, V-shaped central domain (V-domain), and a C-terminal proline-rich domain, whereas the remaining four proteins contain a single Bro1-domain. A BLASTP search using BRAF as a query retrieved no homologs outside the plant kingdom. However, a protein, named BROX, with a single Bro1-domain was also found in mammals but with unknown physiological function[26]. BRAF orthologs can be widely found in eudicots, monocots, and more basal classes of plant. The phylogenetic analysis shows that the plant BRAF proteins can be phylogenetically grouped into distinct classes and there seems to be a unique plant Bro1-domain protein with proposed functions distinct from animal and yeast Bro1-domain proteins (Fig. 2b). Interestingly, sequence alignment of the plant BRAF orthologs reveals that the A330V substitution identified in *sof524* affects an

invariant residue in the Bro1-domain (Supplementary Fig. 3C). Hierarchical clustering of tissue-specific expression patterns of *Arabidopsis* single Bro1-domain genes (without AtBRO1/ALIX) and FREE1 showed the nearest cluster between BRAF and FREE1, indicating their related function in plant development (Supplementary Fig. 3D).

To demonstrate that the point mutation identified in BRAF, hereafter called "*braf-1*", is responsible for the survival phenotype of *sof524*, we performed complementation experiments by transforming *sof524* plants with *BRAF* under its native promoter (*BRAFpro::BRAF*) and the homozygous T3 seedlings showed a lethal phenotype in the DEX treatment (Fig. 2c and Supplementary Fig. 4A). Moreover, the seedling phenotype of *sof524* could also be complemented by *BRAFpro::BRAF-YFP* but not by *BRAFpro::BRAF(A330V)*. In addition, we collected T-DNA insertion mutants *braf-2* and *braf-3* (Supplementary Fig. 4B), which affect the BRAF expression to different degrees as confirmed by quantitative reverse transcription-PCR (qRT-PCR) assay and immunoblotting with a BRAF antibody, respectively (Fig. 2d, e and Supplementary Fig. 4C). Homozygotes of these T-DNA mutants exhibited no noticeable phenotypic differences from the WT under optimal conditions. Similar to *sof524*, 7-day-old seedling of double-homozygous *braf-2 FREE1-RNAi* showed survival phenotype in DEX treatment, while *braf-3 FREE1-RNAi* seedling also partially reverted the lethal phenotype in DEX treatment after 10 days' culture (Fig. 2f). Moreover, seedlings that showed lethal phenotype in the DEX treatment also accumulated ubiquitinated membrane proteins at a higher level than those that showed the survival phenotype (Fig. 2g). Thus, all these genetic complementation and recapitulation data demonstrate that the loss of function of BRAF is responsible for the recovery of the *FREE1-RNAi* lethal phenotype.

We further want to know whether the recovery phenotype is specific to FREE1 or related to the whole ESCRT pathway. Sucrose nonfermenting 7 (SNF7) is an ESCRT-III component and the AAA ATPase Vps4/suppressor of K1 transport growth defect 1 (Vps4/SKD1) is an ESCRT-III-associated protein. Dominant negative mutants SNF7.1(L22W) and SKD1(E232Q) displayed severe cotyledon developmental defects[27]. We transformed the DEX-inducible SNF7.1(L22W) or SKD1(E232Q) into the *braf-2* plants, but T2 plants with transgene showed no recovery phenotype (Supplementary Fig. 4D). In addition, *braf-2* also did not rescue the DEX-inducible *AtBRO1/ALIX-RNAi* and *SH3P2-RNAi* plants (Supplementary Fig. 4D), which are FREE1-associated proteins[15,25]. To our surprise, we also failed to identify a survival double-homozygous line in 86 progenies of

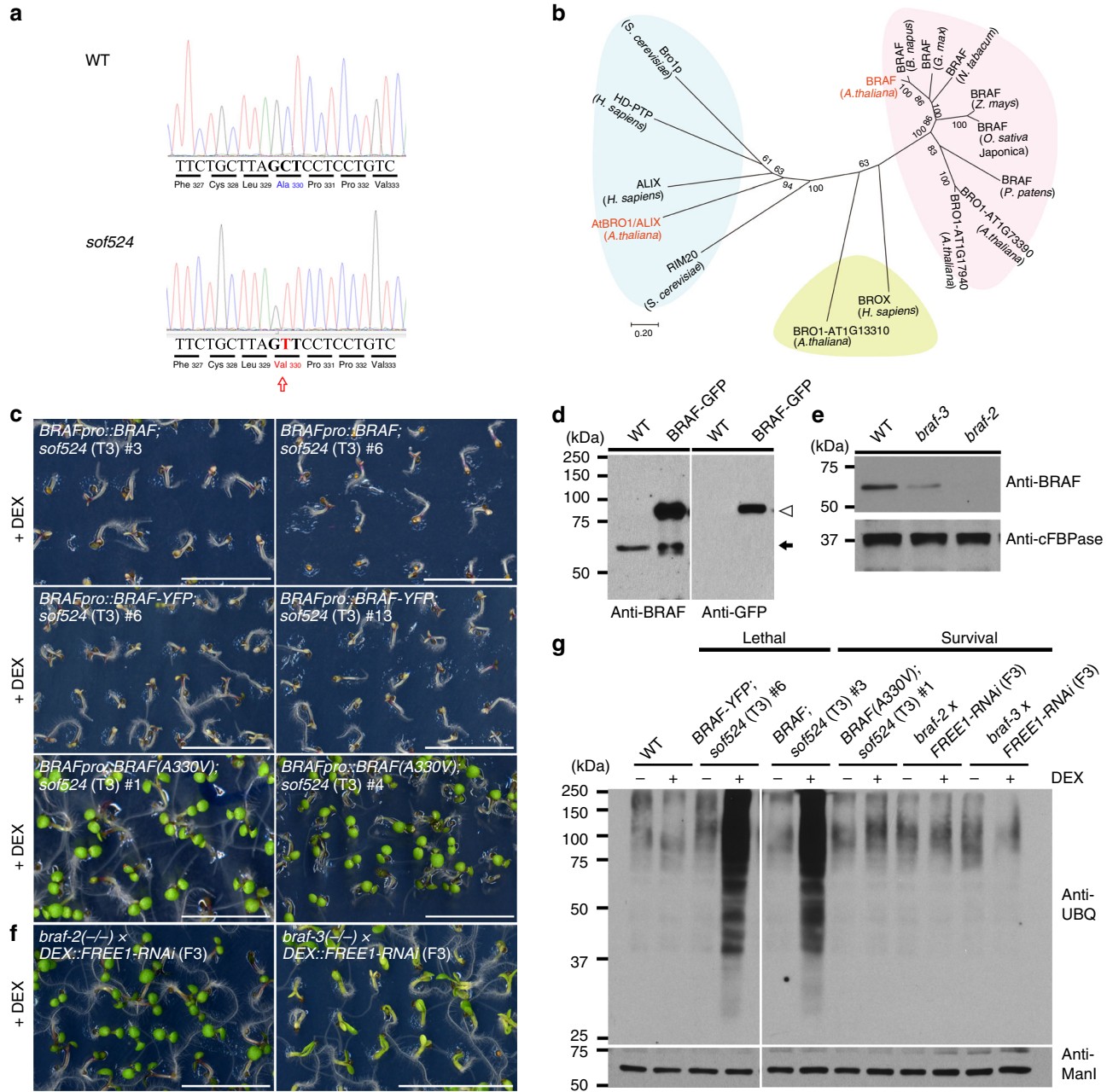

**Fig. 2** The *sof524* mutation affects BRAF. **a** Chromatograms of BRAF open reading frame sequence (Sanger sequencing of PCR-amplified DNA) from WT (top) or *sof524* (bottom). Arrow indicates the identified nucleotide mutation (C-to-T transition), as well as the resulting amino-acid change (alanine[330]-to-valine) in NGS mapping analysis. **b** Neighbor-joining phylogenetic tree, generated with the amino-acid sequences of Bro1-domain-containing proteins, shows the predicted relationship of the Bro1-domain proteins in plants, yeast, and human. Major groupings are indicated by color codes. Note that AtBRO1/ALIX and BRAF group separately. **c** Complementation of *sof524* with *BRAFpro::BRAF*, *BRAFpro::BRAF-YFP*, or *BRAFpro::BRAF(A330V)*. The phenotype of 7-day-old *sof524* seedling expressing indicated proteins growth on MS plates supplied with (+) DEX. Two transgenic lines are shown. Scale bar, 1 cm. **d** BRAF antibody can specifically detect endogenous BRAF (arrow) and BRAF-GFP (arrowhead) fusion proteins. **e** BRAF expression analysis of T-DNA insertion mutants. Seven-day-old seedlings of indicated genotypes were subjected to immunoblot analysis with indicated antibodies. Anti-cFBPase is used as a loading control. **f** The phenotype of 7-day-old seedlings of double-homozygous mutants *braf-2(−/−) DEX::FREE1-RNAi* and 10-day-old of *braf-3(−/−) DEX:: FREE1-RNAi* growth on MS plates supplied with (+) DEX. Scale bar, 1 cm. **g** Immunoblot analysis with anti-UBQ on membrane protein extracts from the indicated seedlings of genotypes. Anti-ManI is used as a loading control

*braf-2(−/−) free1(+/−)*. The possible explanation for the difference might be due to the multiple function of the FREE1 protein in different plant growth stages and suppressor screening using *DEX::FREE1-RNAi* as starting material focuses on the FREE1 function in post-germination growth. All of these results suggested that BRAF specifically converts the lethal phenotype of *FREE1-RNAi* plants.

**BRAF colocalizes with FREE1 and interacts with Vps23**. To get insights on BRAF function in plants and how it functions with FREE1, we first analyzed the subcellular localization of BRAF. To this aim, we used the *Arabidopsis* transgenic lines expressing GFP- or monomeric red fluorescent protein (mRFP)-tagged forms of BRAF, which fully colocalized at the PM and intracellular punctate dots (Supplementary Fig. 5A). BRAF-GFP showed

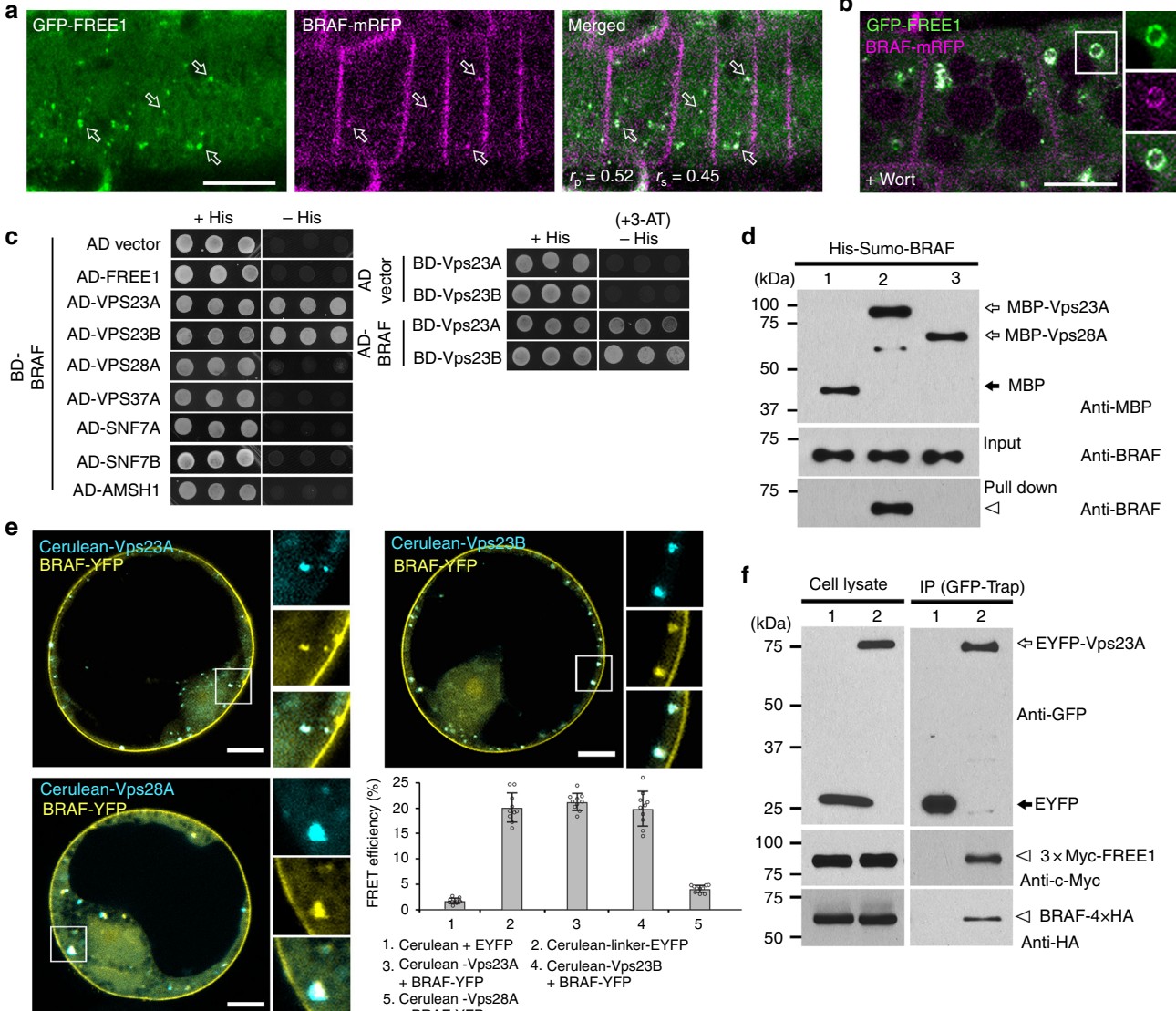

**Fig. 3** BRAF colocalizes with FREE1 in MVB/PVCs and interacts with ESCRT-I component Vps23. **a** BRAF-mRFP shows plasma membrane and intracellular punctate dots. BRAF-mRFP-labeled punctae partially colocalize with GFP-FREE1 in *Arabidopsis* root epidermal cells. Arrows indicate colocalized dots. Colocalization relationship was calculated by Pearson–Spearman correlation. Scale bar, 10 μm. **b** GFP-FREE1 and BRAF-mRFP colocalize to wortmannin (Wort)-induced enlarged MVB/PVCs. Seedlings expressing GFP-FREE1 and BRAF-mRFP as shown in (**a**) were treated with Wort for 40 min before imaging. Separated images of each channel in the white outline area are shown on the right side (from top to bottom: GFP, RFP, and merged). Scale bar, 10 μm. **c** Y2H analysis of the binary interactions of BRAF with FREE1, ESCRT-I component (Vps23A, Vps23B, Vps28A, and Vps37A), ESCRT-III component (SNF7A and SNF7B), or deubiquiting enzyme AMSH1. Transformed yeast cells were grown on either synthetic complete medium lacking leucine and tryptophan (with histidine, +His) as a transformation control, or synthetic complete medium lacking leucine, tryptophan, and histidine (without histidine, −His) for interaction assays. 3-AT is used to suppress the background self-activation of the BD genes. **d** In vitro binding assays of recombinant MBP (lane 1), MBP-Vps23A (lane 2), or MBP-Vps28A (lane 3) with BRAF. Anti-BRAF and anti-MBP antibodies were used to detect bead-retained material. Arrowhead indicates BRAF protein pulled down by Vps23A. **e** FRET analysis of the colocalized punctae between BRAF-YFP and the two Vps23 homologs (Cerulean-Vps23A and Cerulean-Vps23B) or Cerulean-Vps28A. FRET efficiency was quantified by using the acceptor photobleaching approach on the right bottom. For each group, 10 individual protoplasts were used for FRET efficiency quantification and statistical analysis. Error bars are the S.D. of FRET efficiency. **f** Immunoprecipitation (IP) assay shows association between BRAF and FREE1 with Vps23A. *Arabidopsis* protoplasts expressing EYFP (lane 1) or EYFP-Vps23A (lane 2) with 3 × Myc-FREE1 and BRAF-4 × HA were subjected to protein extraction and IP with GFP-trap followed by immunoblot with indicated antibodies. Arrowhead indicates BRAF and FREE1 proteins immunoprecipitated by Vps23A

the PM signals colocalized with the PM labeled fluorescent dye FM4-64, but not with the tonoplast marker mCherry-VAMP711 (Supplementary Fig. 5B, C). BRAF-GFP also showed some intracellular punctate dots, which mainly colocalized with the MVB/PVC marker mCherry-Rha1, but separated from the Golgi marker mCherry-SYP32 and the *trans*-Golgi network/early endosome (TGN/EE) marker VHA-a1-RFP (Supplementary

Fig. 5D–F), indicating that BRAF-GFP punctae represent mostly late endosomes (MVB/PVCs). FREE1 localized to the MVB/PVCs[12]. If BRAF localizes to MVB/PVCs, it should show overlapping intracellular localizations. Indeed, BRAF-mRFP substantially colocalized with GFP-FREE1 in intracellular punctate dots (Fig. 3a). To further establish MVB/PVC localization of BRAF, we treated the GFP-FREE1/BRAF-mRFP co-expressing

line with the PI3K/PI4K inhibitor wortmannin (Wort), which induced MVB/PVC homotypic fusion and resulted in enlarged MVB/PVCs[28,29]. In Wort-treated cells, GFP-FREE1 and BRAF-mRFP showed colocalization to the surface of the enlarged MVB/PVCs that appeared as ring-like structures (Fig. 3b). Collectively, these results demonstrate that BRAF colocalizes with FREE1 in MVB/PVCs.

To identify interacting proteins of BRAF, we performed a yeast two-hybrid (Y2H) screen using the Universal *Arabidopsis* Normalized Library and BRAF as the bait. In this screen, we identified an ESCRT-I component Vps23A (also termed ELC) as an Y2H interactor of BRAF. To explore the possible molecular links of BRAF in plant ESCRT machineries, we next cloned other key subunits of plant ESCRT complexes for Y2H binary assays and found that BRAF interacted with both *Arabidopsis* Vps23 homologs Vps23A and Vps23B, but neither with FREE1 nor other ESCRT-I complex components Vps28A and Vps37A (Fig. 3c). In contrast to *Arabidopsis* AtBRO1/ALIX, which interacts with the ESCRT-III component SNF7/CHMP4 and the de-ubiquitinating enzyme AMSH1[22,23], BRAF did not interact with SNF7, nor with AMSH1 (Fig. 3c). In addition, we did not observe any interaction between Vps23 homologs and the other three single-Bro1-domain proteins in the Y2H binary assays (Supplementary Fig. 6A). All these data point to a unique and specific interaction between BRAF and Vps23.

To further analyze the interaction between BRAF and Vps23, we purified recombinant maltose-binding protein (MBP), MBP-Vps28A, MBP-Vps23A, and Sumo-tagged BRAF (Supplementary Fig. 6B). In an in vitro binding assay, MBP-Vps23A bound BRAF, but not with the controls MBP or MBP-Vps28A (Fig. 3d), indicating that Vps23A and BRAF can directly interact with each other. We further confirmed the specific interaction of BRAF and Vps23 in planta by immunoprecipitation (IP) between BRAF-YFP and Myc-tagged Vps23A or Vps23B with controls, including ESCRT-I (Vps28A and Vps37A) and ESCRT-III components (SNF7A and SNF7B), as well as by acceptor photobleaching fluorescence resonance energy transfer (FRET-AB) analysis between Cerulean-Vps23A or Vps23B and BRAF-YFP (Fig. 3e and Supplementary Fig. 6C). In addition, the IP assay using the protein extracts from *Arabidopsis* cells co-expressing Myc-tagged FREE1, hemagglutinin (HA)-tagged BRAF, and enhanced yellow fluorescent protein (EYFP)-tagged Vps23A showed that both 3 × Myc-FREE1 and BRAF-4 × HA were immunoprecipitated by EYFP-Vps23A, indicating the association of these three proteins in the same ESCRT-I complex (Fig. 3f). Thus, we demonstrated that BRAF was incorporated into the ESCRT-I complex via a direct interaction with Vps23.

**BRAF competes with FREE1 in binding to Vps23**. The Pfam database showed that Vps23 has an N-terminal ubiquitin E2 variant (UEV) domain followed by a proline-rich region (PRR) and coiled-coil (CC) region, and a C-terminal steadiness box (SB) domain (Fig. 4a). To clarify the domain(s) in Vps23 that is(are) required for interaction with FREE1 or BRAF, we next generated different Vps23A fragments for Y2H interaction analysis. As shown in Fig. 4b, the UEV, PRR, and the C-terminal SB region were dispensable for the interaction with FREE1, whereas the CC domain was sufficient for the interaction with FREE1. Similar results also showed that deletion of the CC domain of BRAF but not the N-terminal UEV, PRR, nor C-terminal SB domain abolished their interaction with the Vps23A. In addition, an in vitro binding assay using recombinant MBP, MBP-Vps23A (CC), and Sumo-tagged BRAF or FREE1 demonstrated that both BRAF and FREE1 can interact directly with the Vps23A CC domain (Fig. 4c). Collectively, these results prove that FREE1 and

BRAF interact with Vps23 via the same middle CC domain. Thus, the Vps23 CC domain can serve as an interacting surface for both FREE1 and BRAF binding. If this were true, FREE1 and BRAF should compete for the binding to Vps23. To test this hypothesis, we conducted an in vitro binding competition assay with purified MBP, MBP-Vps23A, FREE1, and BRAF. Relative quantification analysis of the pull-down efficiency in the competition assay demonstrated that an increasing amount of FREE1 in the binding assay lead to a reduction of BRAF binding to MBP-Vps23A, and conversely increasing the BRAF decreased the binding of FREE1 (Fig. 4d). Interestingly, compared to the WT form, BRAF(A330V) containing the *sof524* mutation has a lower competition ability with FREE1 in binding to Vps23A. Such competition binding between FREE1 and BRAF was also confirmed when using the Vps23A CC domain as the bait protein (Supplementary Fig. 7A, B). In addition, both the in vitro pull-down assay and the in vivo IP assay showed that BRAF(A330V), compared to the WT version, displayed reduced interaction with Vps23A (Fig. 4e–g). Taken together, these results indicate that BRAF competes with FREE1 for binding to Vps23 and that the alanine-to-valine mutation in *sof524* decreased the competition binding.

**BRAF depletion increases FREE1 recruitment to MVB membrane**. Based on the in vitro competition assay, depletion of BRAF in plants probably increases FREE1 recruitment to endosomes. To test this hypothesis, we analyzed the localization and the number of GFP-FREE1 punctae in WT and *braf-2* plants. As shown in Fig. 5a and Supplementary Fig. 8A, the GFP-FREE1 signals show slightly nucleus and cytosolic patterns in addition to the intracellular punctae in WT plants. Quantitative analysis from Z-stacking sections showed a significant ($P < 0.01$) increase of GFP-FREE1 punctate dots within the cytosol compared to WT cells. These GFP-FREE1 punctae were localized to MVB/PVCs by performing immunofluorescence labeling using anti-VSR (Fig. 5b), and the colocalization percentage of anti-VSR-labeled MVB/PVCs with GFP-FREE1 vesicles was significantly increased ($P < 0.05$) in *braf-2* (71.4 ± 5.6%, $n = 1265$) compared to WT (58.5 ± 6.5%, $n = 1245$). Similar to the observations in the WT control, these GFP-FREE1 punctae in the *braf-2* also formed enlarged ring-like structures after wortmannin treatment but did not produce "BFA bodies" after Brefeldin A (BFA) treatment (Supplementary Fig. 8A, B). Similar defects were also observed in four individual GFP-FREE1 *braf-2* transgenic lines. To test the possible effect of BRAF depletion on the localization of other organelle markers, we crossed the MVB/PVC marker YFP-Rha1 and the TGN/EE marker VHA-a1-GFP into the *braf-2* plants. Further quantification analysis of the three-dimensional (3D) confocal images showed that either YFP-Rha1 or VHA-a1-GFP exhibited identical patterns with similar numbers of punctae in WT and *braf-2* mutant (Supplementary Fig. 8C–E). These results indicate that the increased recruitment of GFP-FREE1 to MVB/PVCs is specifically caused by depletion of BRAF.

Such a disturbance in FREE1 membrane distribution can also be detected in the DEX-treated *sof524* plants when probed for endogenous FREE1 distribution in cellular fractions. Distinct from WT and *FREE1-RNAi* plants, the RNAi-decreased FREE1 protein in the *sof524* mutant was mainly distributed in the cell membrane (CM) fraction while the distribution of BRAF(A330V) mutant in the cell soluble (CS) fraction was increased (Fig. 5c). Following the logic of competition, more BRAF proteins would be associated with the MVB/PVC membrane in the *FREE1-RNAi* plants than that in the WT. However, BRAF was only slightly increased in the CM fraction of *FREE1-RNAi* vs. WT (Fig. 5c). The possible explanation is that such expected increase in membrane association could be masked by the presence of BRAF

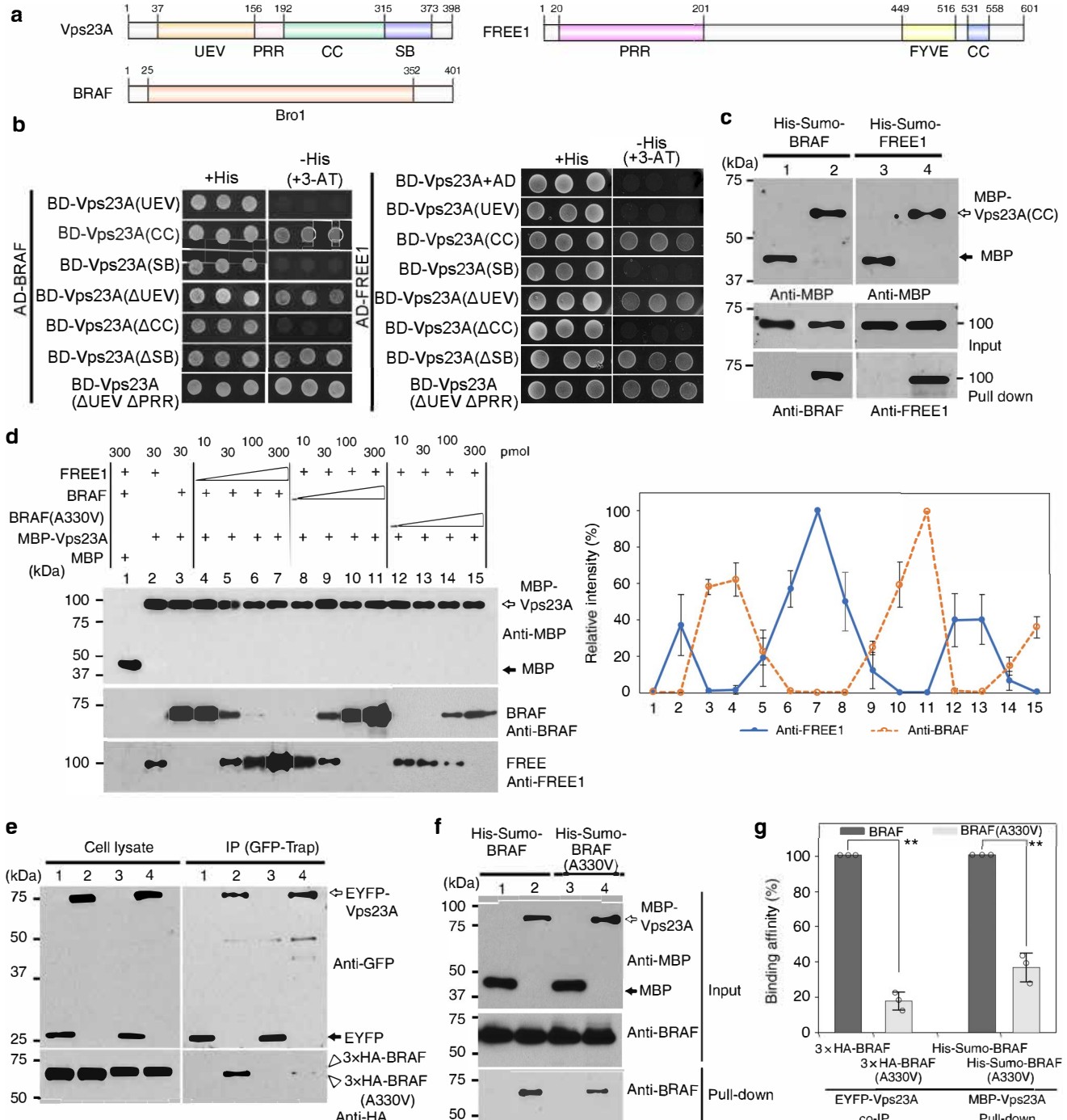

**Fig. 4** BRAF competes with FREE1 for ESCRT-I component Vps23 binding. **a** Domain structures of BRAF, FREE1, and Vps23A. UEV ubiquitin E2 variant, PRR proline-rich region, CC coiled-coil, SB steadiness box. **b** Y2H analysis of the binary interactions between BRAF (left) or FREE1 (right) with different domain deletions of Vps23A. **c** In vitro binding assays of recombinant BRAF (lanes 1 and 2) or FREE1 (lanes 3 and 4) with MBP (lanes 1 and 3) and MBP-Vps23A (CC) (lanes 2 and 4). **d** Competition assay. Binding assay of MBP-Vps23A with 30 pmol of BRAF was performed in the presence of 10, 30, 100, or 300 pmol of FREE1. Binding assay of MBP-Vps23A with constant FREE1 was also performed in the increasing amounts of BRAF or BRAF(A330V). The pull-down efficiency is shown on the right. The amount of proteins on the sepharose was compared with that of the control pair (as 100%) of MBP-Vps23A/FREE1 (lane 7) or MBP-Vps23A/BRAF (lane 11). The intensity of the pull down was normalized by the MBP-Vps23A intensity. Error bars are the S.D. from three independent experiments. **e** Immunoprecipitation (IP) analysis between 3 × HA-BRAF or 3 × HA-BRAF(A330V) and Vps23A. Arabidopsis protoplasts expressing EYFP (lanes 1 and 3) or EYFP-Vps23A (lanes 2 and 4) with 3 × HA-BRAF (lanes 1 and 2) or 3 × HA-BRAF(A330V) (lanes 3 and 4) were subjected to protein extraction, followed by immunoprecipitation (IP) using GFP-Trap agarose beads and subsequent immunoblot analysis on eluted proteins with indicated antibodies. **f** In vitro pull-down assays of BRAF or BRAF(A330V) with Vps23A. Recombinant His-Sumo-BRAF (lanes 1 and 2) or His-Sumo-BRAF(A330V) (lanes 3 and 4) was incubated with either MBP (lanes 1 and 3) or MBP-Vps23A (lanes 2 and 4) for 1 h at 4 °C and subjected to immunoblot analysis. **g** Quantification analysis of binding efficiency in IP analysis and in vitro pull-down assays. The amount of binding protein in BRAF (A330V)-Vps23A was compared with that of the control pair of BRAF-Vps23A (as 100%). The intensity of the immunoblot was normalized by the input. Error bars represent the S.D. from three independent immunoblot results. **P < 0.01 in Student's t-test

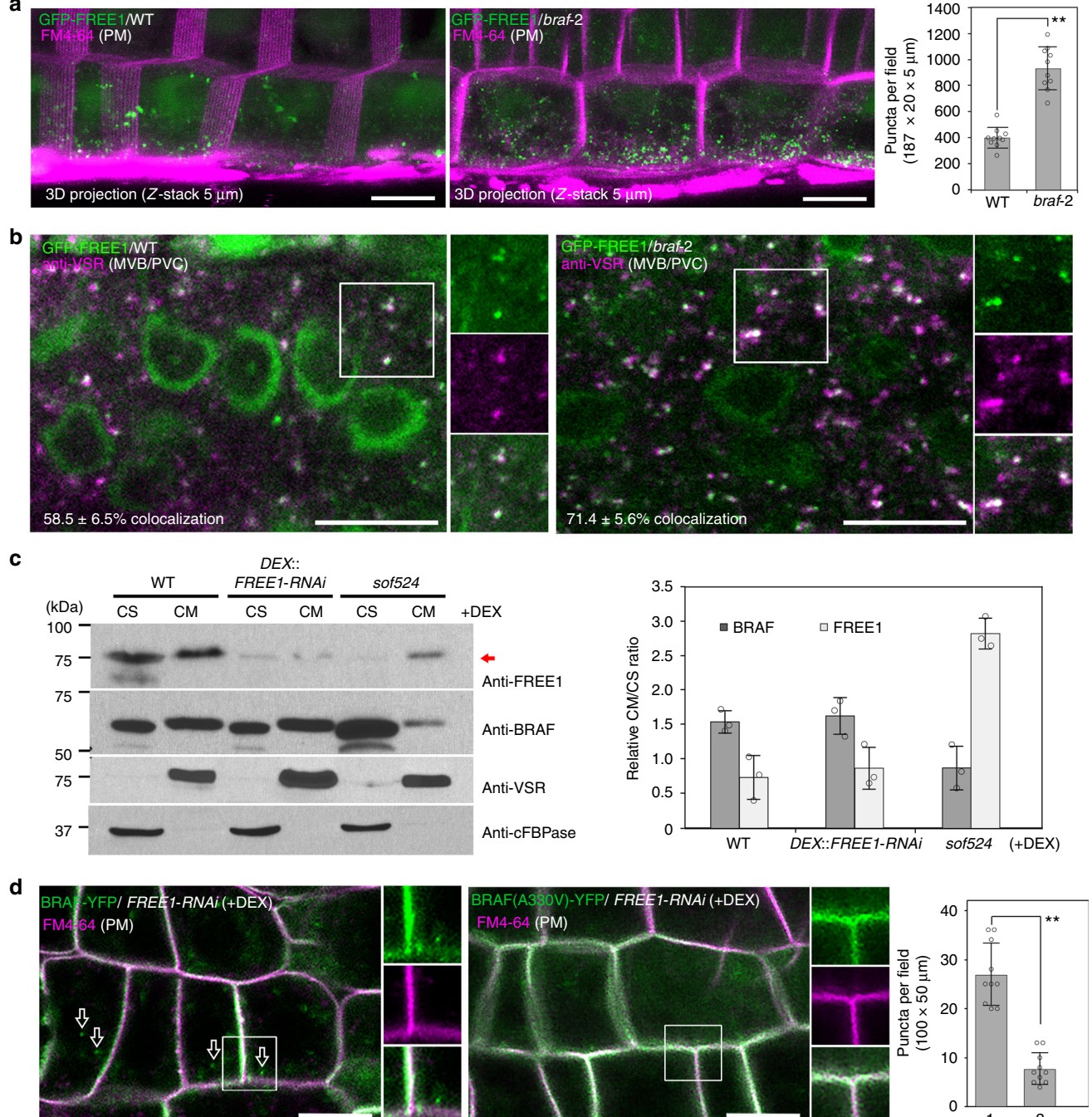

**Fig. 5** BRAF regulates FREE1 recruitment to MVB/PVCs. **a** The number of GFP-FREE1 punctae in *braf* mutant was increased compared to WT. Confocal images were collected from the root epidermal cells of the basal meristem region. Analysis of the number of GFP-FREE1 punctae per root section by *Z*-stack projection is quantified on the right. Ten slices were collected in a total thickness of 5 μm for generating the 3D projection image. FM4-64 was used to label and visualize the cell plasma membranes (PM). The results were obtained from 10 individual seedlings. Error bars represent the S.D. of puncta per field. **P < 0.01 in Student's *t*-test. Scale bars, 10 μm. **b** The GFP-FREE1 punctae in *braf* mutant localizes to MVB/PVCs. Immunofluorescent labeling with the MVB/PVC marker anti-VSR antibody in WT or *braf-2* mutant expressing GFP-FREE1. Colocalization was quantified from five individual labeling roots. The percentage of anti-VSR-labeled MVB/PVCs with GFP-FREE1 vesicle colocalization is included in the bottom. Total numbers of vesicles counted were *n* = 1245 in WT, and *n* = 1265 in *braf-2* mutant. Scale bars, 10 μm. **c** The *sof524* mutation enhanced FREE1 cell membrane distribution. Immunoblot and quantification analysis of the FREE1 and BRAF membrane association in WT, *DEX::FREE1-RNAi*, and *sof524* plants. The FREE1 or BRAF intensity was normalized by the loading control of anti-VSR as cell membrane (CM) fraction and anti-cFBPase as cell soluble (CS) fraction, and the CM/CS ratio of FREE1 or BRAF in indicated genotype seedlings was quantified on the right. Note the increased FREE1 distribution in CM fraction of *sof524* seedlings as indicated by the arrow. Error bars are the S.D. from three independent experiments. **d** The BRAF(A330V) mutation disturbs its membrane association with the endosome, but not the PM. Confocal images were taken from root epidermal cells of 7-day-old seedlings of indicated genotypes after DEX induction. The numbers of intracellular punctae per root section of BRAF-YFP (column 1) or BRAF(A330V)-YFP (column 2) are quantified on the right. The results were obtained from 10 individual seedlings. FM4-64 was used as a plasma membrane (PM) marker. **P < 0.01 in Student's *t*-test. Scale bars, 10 μm

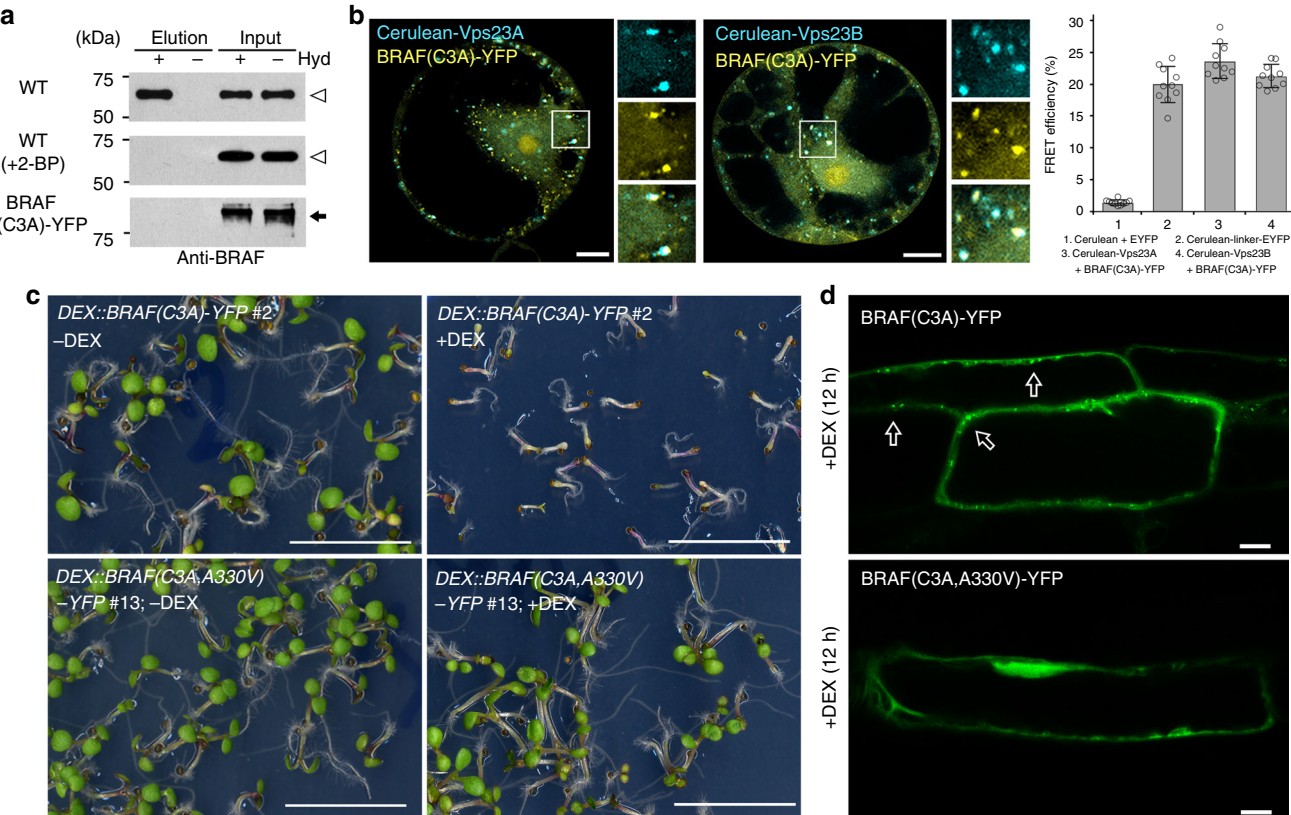

**Fig. 6** BRAF is *S*-acylated and the de-acylated BRAF(C3A)-overexpressing mutant seedling is lethal. **a** Biotin switch assaying the *S*-acylation of BRAF proteins. Total membranes were purified from *Arabidopsis* protoplasts and subjected to the biotin switch assay. Samples were treated with (+) or without (−) the thioester cleavage reagent hydroxylamine (Hyd). The loading controls (Input) show that equal amounts of BRAF were loaded onto neutravidin beads. The lanes "Elution" show the degree of BRAF recovered from neutravidin beads. Arrowhead indicates endogenous BRAF protein. Protoplasts without transformation were treated with 10 μM 2-BP for 24 h before assay (middle panel). BRAF(C3A)-YFP (arrow) was transiently expressed in the *Arabidopsis* protoplasts (low panel). **b** FRET analysis of the colocalized punctae between BRAF(C3A)-YFP and the two Vps23 homologs (Cerulean-Vps23A and Cerulean-Vps23B) showed interactions. FRET efficiency was quantified by using the acceptor photobleaching approach on the right. For each group, 10 individual protoplasts were used for FRET efficiency quantification and statistical analysis. Error bars are the S.D. of FRET efficiency. **c** Seedlings expressing *DEX::BRAF(C3A)-YFP*, but not *DEX::BRAF(C3A,A330V)-YFP*, are lethal after DEX incubation. The phenotype of 7-day-old seedlings of indicated genotypes growth on MS plates supplied with (+) or without (–) DEX. Scale bar, 1 cm. **d** Confocal imaging analysis of the differential intracellular distribution of BRAF (C3A)-YFP and BRAF(C3A,A330V)-YFP in *Arabidopsis* root cells. Six-day-old transgenic *Arabidopsis* plants were transferred into liquid medium containing 30 μM DEX for 12 h to induce the expression of YFP fusions. BRAF(C3A)-YFP are predominantly found in vesicles, while BRAF (C3A,A330V)-YFP signals shown in cytosol. Scale bar, 10 μm

in the PM. Moreover, in DEX-treated *DEX::FREE1-RNAi* plants, BRAF-YFP signals were mainly localized at both PM and intracellular punctae, while the BRAF(A330V)-YFP was mainly localized at the PM (Fig. 5d). Taken together, the increased presence of the remaining FREE1 protein in the membrane fraction in DEX-treated *sof524* was likely due to the poor association of BRAF(A330V) with the MVB/PVC membrane.

**Overexpressed endosomal BRAF disturbs FREE1 membrane association**. If FREE1 and BRAF compete for the binding to Vps23, overexpression of endosome-localized BRAF should affect FREE1 membrane distribution, which in turn will show a seedling lethality phenotype. To investigate this possibility, we transformed the overexpressed *UBQPro::BRAF-GFP* construct into the WT plant for phenotypic observation. However, BRAF-GFP overexpression plants did not show seedling lethality (Supplementary Fig. 9A). The possible explanation is that BRAF overexpression did not result in increased endosome-localized BRAF, when compared with the native promoter-driven *BRAFPro::BRAF-YFP* plant (Supplementary Fig. 5G).

Many PM-localized soluble proteins are often modified by *S*-acylation for anchoring to the membrane[30–33]. Upon de-acylation, the proteins are released from the PM to cytoplasm and increased their recruitments to endosomes[30,33,34]. Thus, if BRAF was a *S*-acylated protein for PM localization, de-acylation of the PM-localized BRAFs would release from the PM into cytosol, serving as a pool for endosome recruitment, and thus increase the endosome-localized BRAF.

Indeed, the GPS-Lipid software[35] predicts that BRAF is a *S*-acylated protein with putative modification site at position 3. Further biotin switch *S*-acylation assay confirmed that BRAF is *S*-acylated (Fig. 6a). Treatment with the protein *S*-acylation inhibitor 2-bromopalmitate or mutation by substituting the predicted *S*-acylation site cysteine residue with alanine, BRAF (C3A)-YFP, resulted in the de-acylation of BRAF (Fig. 6a). Moreover, the expressed BRAF(C3A)-YFP were mainly localized to Cerulean-Vps23A or Cerulean-Vps23B with an interaction capability as determined by FRET-AB (Fig. 6b). Collectively, these results demonstrated that BRAF is *S*-acylated for its PM localization, and that the de-acylated BRAF(C3A) mutant is mainly recruited to the MVB/PVCs and interacts with Vps23.

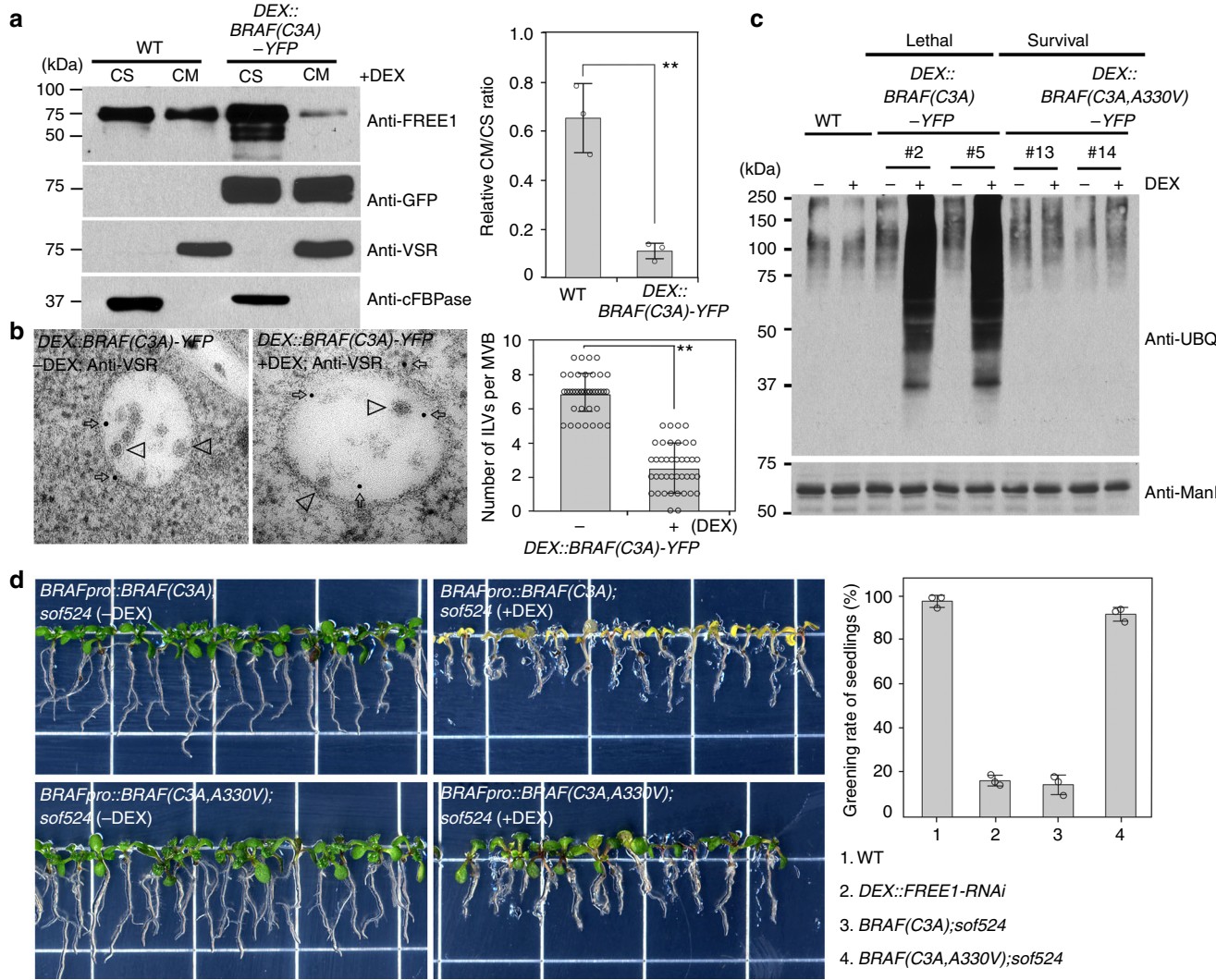

**Fig. 7** BRAF(C3A)-overexpressing mutant causes defects in membrane proteins' vacuolar sorting. **a** Immunoblot analysis of FREE1 membrane association in WT and *DEX::BRAF(C3A)-YFP* plants after DEX induction. The FREE1 intensity was normalized by the loading control of anti-VSR as cell membrane (CM) fraction and anti-cFBPase as cell soluble (CS) fraction, and the CM/CS ratio of FREE1 in indicated genotype seedlings was quantified on the right. Error bars are the S.D. from three independent experiments. **P < 0.01 in Student's *t*-test. **b** Electron micrographs of ILV formation in MVBs. Ultrathin sections were prepared from HPF/FS samples of indicated genotype plant roots without (−) or with (+) DEX induction, followed by immunogold labeling using VSR antibodies. The number of ILVs per MVB was statistically analyzed on 40 MVBs recognized by VSR antibodies. Error bars are the S.D. **P < 0.01 in Student's *t*-test. Scale bar, 100 nm. **c** Immunoblots of membrane protein extracts from 7-day-old seedlings using anti-UBQ or anti-ManI. Note the accumulation of ubiquitin conjugates in *DEX::BRAF(C3A)-YFP* under DEX induction. **d** Complementation of *sof524* with *BRAFpro::BRAF(C3A)* or *BRAFpro::BRAF(C3A,A330V)*. The T1 generation-transformed seedlings were screened on MS with antibiotics for 5 days, and the seedlings were then transferred to MS medium without (−) or with (+) DEX for 7 days. Greening rate of seedlings after DEX treatment was quantified on the right. Error bars are the S.D. from three independent experiments. Scale bar, 1 cm

Next, we used the de-acylated BRAF(C3A) overexpression mutant to investigate the seedling phenotype and possible effect on FREE1 membrane association. We failed to obtain strong fluorescence signals with the *UBQ10pro::BRAF(C3A)-YFP* transgenic line, suggesting that overexpression of BRAF(C3A) could lead to lethality during vegetative plant growth. Indeed, *DEX::BRAF(C3A)-YFP* transgenic lines under DEX treatment showed a seedling lethal phenotype, but not with the lines expressing *DEX::BRAF(C3A,A330V)-YFP*, which harbors the *sof524* mutation (Fig. 6c and Supplementary Fig. 9A). BRAF(C3A)-YFP accumulated in the intracellular compartments in expressing cells (Fig. 6d), which mainly colocalized with the MVB/PVC marker mCherry-Rha1, and partially associated with the Golgi marker mCherry-SYP32, but separated from the TGN/EE marker VHA-a1-RFP (Supplementary Fig. 9B–D), indicating that BRAF(C3A)-

YFP punctae represent mostly MVB/PVCs. By contrast, BRAF (C3A,A330V)-YFP signals were mainly distributed throughout the cytosol (Fig. 6d). To further identify FREE1 membrane distribution in the BRAF(C3A)-YFP-overexpressing plants, CS and CM fractions from WT and BRAF(C3A)-YFP plants were subjected to immunoblotting analysis with anti-FREE1 or anti-BRAF. Interestingly, compared to the WT, an increased amount of FREE1 was found to distribute in the CS fraction with less FREE1 in the CM fraction in plants expressing the BRAF(C3A)-YFP (Fig. 7a). We hypothesize that decrease of the FREE1 on membrane in BRAF(C3A)-YFP mutant would result in failure of ILV formation and consequently accumulation of ubiquitinated membrane cargo. We therefore investigated MVB/PVC morphology at the ultrastructural level. The anti-VSR usually labeled MVB/PVCs with a reduced number of ILVs per MVB section in

DEX-treated BRAF(C3A)-YFP lethal seedlings (Fig. 7b). In addition, high amounts of Ub conjugates with different molecular weight were detected in the membrane fractions isolated from DEX-treated BRAF(C3A)-YFP plants, but not from BRAF(C3A, A330V)-YFP-expressing plants (Fig. 7c). Collectively, these results suggest that overexpression of the endosome-localized BRAF(C3A) mutant disturbs FREE1 membrane association, which affects ILV formation of MVB/PVC and consequently causes accumulation of ubiquitinated membrane cargo.

To further test the hypothesis that dysfunction of endosome-localized BRAF (rather than PM-localized BRAF) in *sof524* induced suppression of lethal phenotype of *FREE-RNAi*, we performed the complementation experiment by transforming endosome-localized *BRAFpro::BRAF(C3A)* and cytosol-localized *BRAFpro::BRAF(C3A,A330V)* into *sof524* plants and observed the plants phenotype with or without DEX treatment. Both *BRAF (C3A); sof524* and *BRAF(C3A,A330V); sof524* plants showed survival phenotype without DEX treatment. After DEX treatment, *BRAF(C3A);sof524* showed seedling lethality phenotype similar to that of *FREE1-RNAi*, which is distinct from the survival phenotype of *BRAF(C3A,A330V);sof524* (Fig. 7d). Taken together, these complementation studies further support that the endosome-localized BRAF functions as FREE1 suppressor.

In conclusion, we demonstrate that the cytosolic BRAF can be recruited to MVB/PVCs via the ESCRT-I component Vps23, or to the PM upon its *S*-acylation in plants. At the MVB/PVC membrane, BRAF regulates FREE1 recruitment to MVB/PVC membrane by their competition binding to the Vps23, and thus function as an important regulator for MVB/PVC function and vacuolar degradation of membrane cargo in plants (Supplementary Fig. 10).

## Discussion

In higher eukaryotes, the essentialness of the ESCRT machinery in regulating ILV formation in MVBs and ubiquitinated membrane protein sorting are well established. However, the existence of regulators modulating the ESCRT machinery in order to control MVB/PVC organelle biogenesis and maintain membrane protein homeostasis remains unclear. To summarize our findings, we first identified a plant Bro1-domain protein that regulates the ILVs of MVB formation by demonstrating (1) through both in vitro and in vivo biochemical analysis that BRAF is incorporated into the ESCRT-I complex and competes with FREE1 for binding to Vps23; (2) with genetic data, which show that depletion of BRAF increases FREE1 recruitment to MVB/PVCs, providing a rationale for the reversion of *FREE1-RNAi* phenotypes; and (3) that overexpression of endosome-localized BRAF gives rise to a seedling lethality phenotype as do FREE1 loss-of-function mutants with defects in the formation of ILVs in MVBs leading to the accumulation of ubiquitinated membrane proteins. These observations would seem to argue that BRAF is a regulator that negatively controls the formation of ILVs in MVBs in plants mainly through competing with the unique plant FREE1 protein for binding to the ESCRT-I component Vps23, an upstream component of the ESCRT pathway. This regulatory mechanism is distinct from the known positive regulators identified in higher eukaryotes, including the evolutionarily conserved LYST-interacting protein 5 (LIP5) and IST1-like 1 (ISTL1) as its animal and yeast counterparts as well as plant-specific positive regulator of SKD1 (PROS), which coordinately regulate SKD1/Vps4 ATPase activity to catalyze the disassembly and recycling of the ESCRT-III components back to the cytoplasm[36–38].

The existence of such a pathway is meaningful, as receptors need to be constitutively turned over and recycled back to the PM for proper cellular functions as well as degradation. For example,

the phytohormones auxin and gibberellin as well as cytokinin mutually coordinate their activities to regulate the abundance and polar distribution of PINs by modulating their endocytosis and endosomal recycling in response to different phytohormone interplay in different plant organs or tissues[39–41]. However, how the ubiquitinated receptors/transporters such as PIN2 are exactly balanced between recycling from MVB/PVCs and being recruited by the ESCRT machinery and engulfed into ILVs for degradation in regard to different hormones needs future investigation[42]. Our study may provide hints on an important regulating mechanism that a factor such as BRAF may modulate ILV formation through controlling the assembly of the ESCRT machinery. This then would prevent access to ubiquitinated membrane proteins for degradation during organogenesis or plant growth.

Multiple studies have shown that certain cytoplasmic proteins, including the mammalian Ras proteins[43], the plant MfNACsa (a NAM/ATAF1/2/CUC2 transcription factor)[44], and the plant LIP1 (Lost In Pollen Tube Guidance 1)[45], are *S*-acylated for anchoring to PM, while the de-acylated proteins are released from the PM to cytosol and increased their recruitments to endomembrane compartments[33,43,44,46]. Based on the findings of this study, we propose a working model of BRAF function in plants (Supplementary Fig. 10) as follows: (1) BRAFs are cytoplasmic pool proteins that can be recruited to MVB/PVCs via interaction with the ESCRT-I components Vps23, or to the PM upon their *S*-acylation; (2) upon their de-acylation, the PM-localized BRAF proteins are released from the PM into the cytoplasm, resulting in increased cytosolic pool of BRAF for MVB/PVCs recruitment; (3) at the MVB/PVC membrane, BRAF competes with FREE1 through direct binding to the ESCRT-I component Vps23, thus negatively regulating FREE1 recruitment to MVB/PVCs; and (4) the *sof524* mutant contains BRAF(A330V) mutation in the *FREE1-RNAi* plants. The BRAF(A330V) has a lower competition ability with FREE1 in binding to Vps23. In the DEX-treated *sof524* mutant plants, the decreased competition ability of BRAF (A330V) leads to the increased recruitment of the RNAi-decreased FREE1 to MVB/PVCs, thus complementing the defects in *FREE1-RNAi* plant. However, it remains unknown if the PM-localized *S*-acylated BRAF would be recruited directly to MVB/PVCs, or to other intracellular compartments (e.g. Golgi) upon its de-acylation and release from the PM. In addition, since the *S*-acylation of proteins could be triggered and regulated by certain environmental cues, e.g. drought stress[44], it is highly speculated that the cells may utilize certain environmental or growth signals to regulate the distributions of BRAF proteins in the cytoplasm, the PM, and the MVB/PVCs by the *S*-acylation process[31–33], which will be an interesting topic for future study.

## Methods

**Plant materials**. The *Arabidopsis* T-DNA insertion line of *braf-2* allele (SALK_145102) and *braf-3* allele (GK-134H01) was obtained from NASC and GABI-Kat, respectively. The single T-DNA insertion in individual line was confirmed using the genome-sequencing data by strategy to identify chimeric reads with both T-DNA and genome DNA sequences[17]. Homozygous mutants of the expected genotypes were selected by PCR using gene- and T-DNA-specific primers. To generate the transgenic plants, all of the resulting constructs were introduced into *Agrobacterium* and transformed into WT Col-0 or indicated mutants by floral dip[47]. *Arabidopsis* WAVE lines expressing fluorescent protein-tagged organelle markers were obtained from NASC.

**Plasmid construction**. For the constructs used for transient expression in *Arabidopsis* protoplasts, the cDNAs encoding the corresponding genes were amplified and cloned into pBI221 vectors modified to containing Cerulean, GFP, EYFP, and YFP under the UBQ10 promoter by restriction digestion. For fluorescent protein-tagged BRAF transgenic plant, coding sequence without stop codon was amplified from cDNA and recombined using Gateway cloning (Invitrogen) into the pDONR/Zeo vector, which was further cloned into PCAMBIA1300-UBQpro::GW-mRFP or PBI121-UBQpro::GW-GFP vectors modified to containing mRFP or GFP under the UBQ10 promoter. For *BRAFpro::BRAF-YFP* transgenic plant, BRAF was cloned

into pBI221-YFP backbone to construct BRAF-YFP, the BRAF promoter region was then fused with BRAF-YFP by PCR fusion and cloned into pDONR/Zeo for further cloned into PGWB vector for plant transformation. BRAFpro::BRAF (A330V)-YFP, BRAFpro::BRAF, BRAFpro::BRAF(A330V), BRAFpro::BRAF(C3A), and BRAFpro::BRAF(C3A,A330V) were cloned using the same strategy. For *DEX:: BRAF(C3A)-YFP* and *DEX::BRAF(C3A,A330V)-YFP* transgenic plant, BRAF(C3) was cloned into pBI221-YFP backbone, the coding region was then cloned into pDONR/Zeo for further cloned into pTA7002-DEX-GW vector for plant transformation. For Y2H analysis, the cDNAs were cloned into pGBKT7 and pGADT7 vectors. The vectors harboring key subunits of plant ESCRT complexes have been previously described[11]. For expression of recombinant proteins, the corresponding cDNAs were cloned into in-house pHisSUMO or pRHisMBP vectors for construction of His-Sumo or MBP fusions, respectively[12,48]. All constructs were confirmed by Sanger sequencing. Primers used for plasmid construction, genotyping, or qRT-PCR are listed in Supplementary Table 1.

**Plant growth and chemical treatment.** Seeds were surface-sterilized and grown on plates with full MS salts (pH 5.7) plus 3% sucrose and 0.8% agar at 22 °C under a long day (16 h light/8 h dark) photoperiod. For treatment with DEX, 10 µmol/L DEX (30 mmol/L stock dissolved in ethanol) was added in the medium. For DEX induction in liquid medium, seedlings were transferred in liquid MS with ethanol as control or 10 µmol/L DEX for indicated time before confocal laser scanning microscopy observation or protein extraction. For boron treatment, 5-day-old *BOR1-GFP* seedcgqtedium containing 0.3 µM boric acid (−B) or 100 µM boric acid (+B) as previously described[49]. FM4-64 uptake experiments were performed by incubation with FM4-64 (12 µM) for 5 min or 6 h, and washed twice with MS medium before observation. BFA and wortmannin were prepared in dimethyl sulfoxide and used at 10 µM for 2 h and 33 µM for 40 min, respectively, in the liquid medium.

***Arabidopsis FREE1-RNAi* suppressor screening.** The mutant pool was established in previous report[17]. Basically, seeds of the *Arabidopsis thaliana* (Col-0) expressing single insertion of pTA7002-FREE1-RNAi were mutagenized by EMS. Approximately, 40 000 seeds corresponding to the progeny of 10 000 mutagenized M1 seeds were sown on MS plates supplemented with 10 µM DEX and were grown for 5 days. Seedlings showing survival phenotype were selected as Suppressors of *FREE1* (*sof*). Selected M2 seedlings were planted into soil for individual M3 seeds collection. Individual M3 seeds were screened on MS medium plus with 10 µM DEX and hygromycin again, and 7-day-old M3 seedlings were screened for survived phenotype.

**Identification of the *sof524* mutation.** The *sof524* mutant (Col-0) was firstly outcrossed with the Landsberg erecta (Ler) WT. The genomic DNA was extracted using DNeasy Plant Mini Kit (Qiagen) from selected surviving F2 seedlings on MS plates supplied with DEX and hygromycin. These sheared DNAs were sequentially ligated with the 30 and 50 adapters using the DNA library Preparation Kit (Illumina, USA), and then the libraries were barcoded and sequenced on an Illumina Hiseq2000 to generate 100-bp pair-end reads, yielding >15-fold genome coverage[17,50]. Sequencing reads of *sof524* were aligned against the Col-0 reference genome sequence (TAIR10) using SOAP2 (http://soap.genomics.org.cn/soapaligner.html), and consensus was called using SAMTOOLS program[51]. NGS mapping was carried out based on 461 070 single-nucleotide polymorphism (SNP) markers using SHORE map outcross function[52]. Relatively reliable loci were filtered as below: consensus quality > 20 (error rate, 1%), total depth > 5. Only EMS-induced C/G to T/A transition SNP markers were further considered as candidates.

**Antibodies.** Synthetic peptide (GeneScript), IPEVAFRKSQTYGYLLEEEE-KAMQC, corresponding to the C termius of BRAF were conjugated with keyhole limpet hemocyanin and used to immunize rats at the Laboratory Animal Services Center of The Chinese University of Hong Kong. Antibodies were affinity-purified using CnBr-activated Sepharose 4B (Sigma-Aldrich; Cat. No. C9142) column conjugated with the peptides. FREE1, EMP12, ManI, and VSR antibodies were homemade[12,28,53]. Additional primary antibodies used were anti-cFBPase (Agrisera, AS04 043), anti-Myc (Santa Cruz, SC-789), anti-UB (Santa Cruz, P4D1), anti-MBP (NEB, E8038S), and anti-HA (Abcam, ab18181).

**Plant protein preparation.** For total protein extraction from plants, whole seedlings or different plant tissues were ground in liquid nitrogen and extracted with the above lysis buffer containing 1% SDS. For CS and CM fraction isolation, the 7-day-old *Arabidopsis* seedlings were ground in ice-cold extraction buffer (40 mM HEPES-KOH at pH 7.5, 1 mM EDTA, 10 mM KCl, 0.4 M sucrose, 0.5 mM phenylmethanesulfonyl fluoride, 25 µg/mL leupeptin, and 1× Complete Protease Inhibitor Cocktail). After centrifugation at $600 \times g$ for 3 min to remove large cellular debris, the supernatant was further centrifuged at 100 000 × g for 60 min at 4 °C. The supernatant was assigned as soluble fraction and the pellet was washed twice and assigned as the membrane fraction.

**Immunoblot analysis.** Protein samples were subjected to gel electrophoresis on 10% or 15% SDS-polyacrylamide gel electrophoresis (SDS-PAGE) gels. Proteins were transferred to nitrocellulose membrane membranes (Bio-Rad) followed by blocking in PBS-T (0.05% Tween-20) with 5% milk powder and antibody incubation. Membranes incubated with horseradish peroxidase-conjugated antibodies were developed using Clarity Western ECL substrate solutions (Bio-Rad). Quantification of immunoblots was done using the ImageJ Software. The uncropped western blot images used in this paper can be seen in Supplementary Figures 11–17.

**Immunoprecipitation.** To prepare cell extracts from protoplasts, transformed protoplasts were first diluted with 250 mM NaCl for threefold and then harvested by centrifugation at $100 \times g$ for 5 min, followed by resuspended in lysis buffer containing 10 mM Tris-HCl, pH 7.4, 150 mM NaCl, 0.5 mM EDTA, 0.4% Nonidet P-40, 5% glycerol, 1 mM dithiothreitol, and 1× Complete Protease Inhibitor Cocktail. The protoplasts were further lysed by passing through a 1 mL syringe with needle and then spun at $600 \times g$ for 3 min to remove intact cells and large cellular debris. The supernatant total cell extracts were then centrifuged at 14 000 rpm for 30 min at 4 °C. The supernatant was prepared in IP buffer (10 mM Tris-HCl, pH 7.4, 150 mM NaCl, 0.5 mM EDTA, 0.2% Nonidet P-40, 5% glycerol, 1 mM dithiothreitol, and 1× Complete Protease Inhibitor Cocktail) and were then incubated with GFP-Trap agarose beads (ChromoTek) for 4 h at 4 °C in a top to end rotator. After incubation, the beads were washed four times with ice-cold washing buffer (10 mM Tris-HCl, pH 7.4, 150 mM NaCl, 0.5 mM EDTA, 0.05% Nonidet P-40, and 5% glycerol) and then eluted by boiling in reducing SDS sample buffer[15,54,55]. Samples were separated by SDS-PAGE and analyzed by immunoblot using appropriate antibodies.

**Recombinant protein purification and in vitro binding assay.** His-Sumo-BRAF or His-Sumo-BRAF(A330V) recombinant proteins were expressed in *Escherichia coli* BL21(DE3) pLysS strain upon induction with 0.2 mM isopropyl b-D-1-thio-galactopyranoside (IPTG) at 28 °C overnight, followed by purification using the His Spin Trap Kit (GE). MBP and MBP-Vps23A tagged proteins expressed in *E. coli* BL21(DE3) pLysS strain upon induction with 0.2 mM IPTG at 16 °C overnight, followed by purification using amylose resin (NEB). The purification of FREE1 has been previously described[12]. Eluted proteins were dialyzed against pull-down buffer (50 mM Tris-HCl, 100 mM NaCl, and 10% glycerol, pH 7.5). Protein concentration was determined using Bio-Rad Protein Assay (cat. no. 5000006) and Coomassie Brilliant Blue staining of proteins' SDS-PAGE using the bovine serum albumin as a standard.

For the in vitro pull-down assay, amylose resin was saturated with MBP or MBP-Vps23A. Amylose resin bound with MBP and MBP-Vps23A were then incubated with 30 pmol BRAF or BRAF(A330V) in 200 µL cold pull-down buffer containing 0.2% Triton X-100 for 1 h at 4 °C in a top to end rotator. The beads were then washed three times with cold buffer A containing 0.2% Triton X-100 and 1 mM dithiothreitol and three times more with pull-down buffer. The bound proteins were eluted by boiling in SDS sample buffer, separated by SDS-PAGE for immunoblot using appropriate antibodies. For the competition of BRAF and FREE1, the in vitro binding assay was performed with the addition of 10, 30, 100, or 300 pmol of His-SUMO-FREE1, His-SUMO-BRAF, or His-SUMO-BRAF (A330V) to each reaction with consistent amount of amylose resin bound with MBP-Vps23A.

*Biotin switch assay for S-acylation*: Cell extracts are treated with the sulfhydryl reactive reagent N-ethylmaleimide to block free cysteine residues. The sample is then divided into two equal portions in which one treated with the *S*-acyl group cleavage reagent hydroxylamine (Hyd⁺) and the other one without (Hyd⁻) as control. Both samples are then treated with sulfhydryl reactive biotin. Following biotinylation, a sample of each reaction is collected as a column loading control (input Hyd⁺ and input Hyd⁻). The remaining sample is loaded onto neutravidin beads and biotinylated proteins are captured on a streptavidin column and eluted (elution Hyd⁺ and elution Hyd⁻). Protein eluates (elution Hyd⁺ and elution Hyd⁻) and loading controls (input Hyd⁺ and input Hyd⁻) are then analyzed by western blotting[56].

**Transient expression in *Arabidopsis* protoplasts.** The *Arabidopsis* suspension cultured cells were maintained by subculture every 5 days, and transient expression experiments were performed using *Arabidopsis* protoplasts from PSB-D suspension cultured cells[54,57]. Briefly, the 5-day-old *Arabidopsis* suspension cells were firstly digested with enzyme solution (1% cellulase "ONOZUKA" RS, 0.05% pectinase, and 0.2% driselase from *Basidiomycetes* sp). After three times wash with electroporation buffer (0.4 M sucrose, 2.4 g/L HEPES, 6 g/L KCl, and 600 mg/L CaCl₂·2H₂O, pH 7.2), the protoplasts were transformed with plasmids via electroporation. The protoplasts were then incubated for 10–14 h prior to confocal imaging analysis or protein extraction.

**Immunofluorescence labeling in *Arabidopsis* roots.** The roots of 5-day-old seedlings were fixed with 4% paraformaldehyde in PBS supplement with 0.1% Triton X-100. After procedures of cell wall digestion, permeabilization, and blocking, the roots were incubated with anti-VSR antibody at 4 µg/mL overnight at

4 °C for immunolabeling, and then were probed with Alexa 568 goat anti-rabbit IgG (Invitrogen) secondary antibody for confocal observation[58].

**Confocal microscopy and FRET analysis**. Confocal fluorescence images were acquired using the Leica SP8 laser scanning confocal system with a ×63 water lens. A sequential acquisition was applied when observing these fluorescent proteins. Images were processed using ADOBE PHOTOSHOP software (http://www.adobe.com/). For each experiment or construct, more than 30 individual cells or 10 individual plants were observed for confocal imaging that represented >75% of the samples showing similar expression levels and patterns. The Pearson–Spearman correlation for colocalization relationships was calculated using ImageJ (Wayne Rasband, NIH, https://imagej.nih.gov/) and the PSC plug-in as described previously[59]. Results are presented either as Pearson correlation coefficients or as Spearman's rank correlation coefficients, both of which produce $r$ values in the range $(-1, 1)$, where 0 indicates no discernable correlation and $+1$ and $-1$ indicate strong positive or negative correlations, respectively. To calculate the colocalization percentage of the GFP-FREE1 vesicle number with anti-VSR punctae, the anti-VSR punctae total number and punctae that were colocalized with GFP-FREE1 vesicles were manually counted[16,22]. Colocalization was quantified from five individual labeling roots. To ensure the cortical focal planes were at an identical level, the cell middle layer was chosen defined from FM4-64 staining and DIC images, and 10 slices were collected in a total thickness of 5 µm for the 3D projection image generation and the punctae calculation using ImageJ.

FRET acceptor bleaching analysis was basically conducted on Leica SP8 confocal system according to the manufacturer's instruction. Briefly, protoplasts expressing various Cerulean and YFP fusions were used for photobleaching using 514 nm laser in full power intensity. Cerulean donor fluorescence was imaged before and after bleaching a region of interest of EYFP to <10% of its initial intensity. FRET efficiency was calculated as Ef $= 100 \times$ (Ipost − Ipre)/Ipost, where Ipre and Ipost stand for the Cerulean intensities before and after acceptor bleaching, respectively. At least 10 individual protoplasts were used for FRET efficiency quantification and statistical analysis. Free Cerulean and EYFP were coexpressed as a negative control. For positive control, Cerulean-linker-EYFP fusion, linked by 18 amino-acid peptide, SSSELSGDEVGGTSGSEF, was used.

**TEM study**. For high-pressure freezing, 4-day-old root tips with indicated treatments were cut and immediately frozen in a high-pressure freezer (EM PACT2, Leica, Germany), followed by subsequent freeze substitution in dry acetone containing 0.1% uranyl acetate at −85 °C in an AFS freeze-substitution unit (Leica, Wetzlar, Germany)[60]. Infiltration with HM20, embedding, and ultraviolet polymerization were performed stepwise at −10 °C. Immunogold labeling was performed as described previously with anti-VSR antibody at 40 µg/mL, and gold-coupled secondary antibody at 1:50 dilution. TEM examination was done with a Hitachi H-7650 transmission electron microscope with a charge-coupled devise camera operating at 80 kV (Hitachi High-Technologies Corporation, Japan).

**Y2H analysis**. Y2H analysis was performed using the MatchMaker GAL4 Two-Hybrid System 3 (Clontech) according to the manufacturer's instructions. A Mate and Plate Normalized Arabidopsis Universal Library (Clontech) was used for mating and screening according to the manufacturer's protocols. For Y2H pairs analysis, the cDNAs were cloned into the pGBKT7 and pGADT7 vectors (Clontech), and plasmids of each pairs were cotransformed into the yeast strain AH109. Transformants were selected on synthetic drop-out (S.D.) medium lacking Trp and Leu (SD-Trp-Leu) at 30 °C, while the selection of interactions was conducted on SD medium lacking His, Trp, and Leu (SD-His-Trp-Leu) containing 0–10 mM 3-amino-1,2,4-triazole for 2 days at 30 °C. The experiments were performed at least twice independently, and similar results were obtained.

**Quantitative reverse transcription-PCR**. Total RNA was extracted using TRIzol (Invitrogen) and treated with RQ1 RNase-Free DNase (Promega) to remove residual genomic DNA; first-strand cDNA was synthesized using M-MLV Reverse Transcriptase (Promega); PCR were performed on a CFX96™ Real-Time PCR System with SYB Green supermix (Bio-Rad, CA, USA). The expression level was normalized against UBQ10.

**Phylogenetic analysis**. The Bro1-domain protein sequences were obtained from NCBI and aligned using ClustalX2.1[61]. Evolutionary analyses were conducted in MEGA7[62] using the Neighbor-Joining method[63]. The tree is drawn to scale, with branch lengths in the same units as those of the evolutionary distances used to infer the phylogenetic tree. The evolutionary distances were computed using the Poisson correction method and are in the units of the number of amino-acid substitutions per site.

**Quantification and statistical analysis**. No statistical methods were used to predetermine samples or outcomes. Data were excluded when negative or positive controls were not working. Sample numbers and the number of biological replicates for each experiment are indicated in figure legends or methods section above. Data are presented as mean values ± S.D. Two-tailed Student's $t$-test was used when

data met criteria for parametric analysis (normal distribution or equal variances). Differences in means were considered statistically significant at $P < 0.05$. Significance levels are: *$P < 0.05$; **$P < 0.01$. Experiments were repeated independently at least three times.

**Accession numbers**. The *Arabidopsis* Genome Initiative locus identifiers for the genes mentioned in this article are BRAF (AT5G14020), AtBRO1 (AT1G15130), FREE1 (AT1G20110), Vps23A/ELC (AT3G12400), Vps23B (AT5G13860), Vps28A (AT4G05000), Vps37A (AT3G53120), SNF7A (AT2G19830), SNF7B (AT4G29160), Vps4 (AT2G27600), and UBQ10 (AT4G05320). GenBank accession numbers for proteins used in the phylogenetic analysis are Bro1p (NP_015241.1), RIM20 (NP_014918.1), *Homo sapiens* ALIX (NP_037506.2), *Homo sapiens* HD-PTP (NP_056281.1), *Homo sapiens* BROX (NP_653296.2), *Nicotiana tabacum* BRO1 (XP_016498689.1), *Zea mays* BRO1 (NP_001169451), *Brassica napus* (XP_013740149.1), *Glycine max* (XP_014626926.1), *Oryza sativa* Japonica Group (XP_015615821.1), and *Physcomitrella patens* (XP_001754811.1).

## Data availability

The authors declare that all data supporting the findings of this study are available within the article and its Supplementary Information files, or from the corresponding author upon reasonable request.

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

## Acknowledgements

We thank Lorenzo Frigerio (University of Warwick) for sharing transgenic plants expressing spL-RFP, Jiri Friml (Institute of Science and Technology Austria) for providing with transgenic plant expressing PIN2-GFP, and Roger W. Innes (Indiana University) for providing with pTA7002-DEX-GW backbone. This work was supported by grants from the Research Grants Council of Hong Kong (G-CUHK 403/17, CUHK14130716, 14102417, C4011-14R, C4012-16E, C4002-17G, and AoE/M-05/12), the National Natural Science Foundation of China (31470294 and 31670179), and CUHK Research Committee to L.J.

## Author contributions

J.S., Q.Z., C.G., and L.J. designed the research. J.S., Q.Z., C.G., Y.Z., and X.W. performed experiments. J.S., Y.Z., Y.Z., Q.Z., and L.J. analyzed the data. J.S, Q.Z., and L.J. wrote the manuscript with comments from all authors.

## Additional information

**Competing interests:** The authors declare no competing interests.

