## [Peer Review File · Nature Communications]

Reviewers' comments:

Reviewer #1 (Remarks to the Author):

Shen et al. describes the unique plant Bro1-domain containing protein BRAF that is responsible for the recruitment of FREE1 to the MVB by competing with FREE1 to bind the CC-domain of the ESCRT-I component VPS23. To my knowledge, these findings are completely new. A novel plant ESCRT-I component is introduced and thus this publication provides essential information for people in plant science but also beyond the plant field. The described competitive binding between proteins of the ESCRT-I machinery sheds light on the possibility of a highly transient ESCRT machinery in plants that will influence scientists working with the ESCRT machinery. The experiments are well planned and the results are highly underlined by a huge amount of negative controls. The manuscript is clearly written except the legends: There could be more information included. e.g. Suppl. Fig. 1: what is cFBPase and why was it used as reference protein? Fig. 5, the meaning of CM is described in the method part, but it would be more convenient to have the description in the legend; Suppl. Fig 3; A, it is not clear which light blue bar and what is the meaning of the blue line, and additional bar; Suppl. Fig. 5: which tissue? VAMP711 is marking the tonoplast (not indicated here). Methods are clearly described and Shen et al. provides sufficient methodological details.

I would suggest following additional experiments to strengthen the paper concerning the localization of BRAF at MVBs and to clearly identify the GFP-FREE1 signal:

.) Shen et al. speculates that BRAF localizes at MVBs by showing a co-localization of BRAF and FREE1 (Fig. 3A). Here, an additional experiment would characterize the co-localization, especially, when the BRAF-mRFP is that weak; for example an inhibitor treatment with Wortmannin would clearly define the structure as MVB.

.) Fig. 5: A, B: It seems that there is additional GFP-FREE1 signal in huge ring like structures or bubbles which were not mentioned. What are these structures? There is also some lumen structure in the in vivo pictures (A). Are these artefacts? It would strengthen the in vivo study (A) using FM4 to see if the cells are still alive.

.) Suppl. Fig. 7: A: Again, the GFP-FREE1 signal is accumulated in big bubbles. Is this the nucleus? Vacuole? This hasn't been shown in recent publications and not in Figure 3A. B: it would strengthen the results if FM4 would be used here (like in A) to see the cell membranes and thus use FM4 as viable marker. Is the scale in B really the same as in A? It seems that the cells are bigger in B than in A.

Minor revisions:

page 3: line 11: 'with' should be deleted

page 5: The last conclusion sentence is formulated in a highly negative way: something is shown that this is not affecting something that negatively affects something. I would prefer a positive message here.

page 6: free1 is described with altered ILV formation and causing accumulation of ubiquitinated membrane cargo in endosomes and at the tonoplast. It is thus supposed, that Shen et al. shows here some microscopic data but Suppl. 2A shows a schema to summarize this phenotype. This should be somehow rewritten.

page 6: line 13: I would prefer 'at' the vacuolar membrane. This would be true for the whole manuscript (e.g. page 11, line 14.)

page 9: line 21/22: What is the S in the BRAFSpro.....?

page 21: line 2: 'seedlings' two times

page 27: line 8: Yeast two-hybrid not yeast-two hybrid
Legends need more detailed description.

Thus, I would like to invite the authors to revise their manuscript to address these specific concerns

and I would be very happy to review again the resubmission.

Kind regards,
Verena Ibl

Reviewer #2 (Remarks to the Author):

The manuscript by Shen et al., report a plant unique regulatory mechanisms of MVB/PVC, which is mediated by proteins with Bro1-domain. Authors showed that *sof524* mutant rescues phenotypes of FREE1-RNAi, and responsible gene of *sof524* encodes plant unique type Bro1-domain containing protein, named BRAF. In addition, they showed that BRAF localized on the plasma membrane and FREE1-positive endosomes, and BRAF compete with FREE1 for Vps23 binding on the endosome. Finally, they showed overexpression of endosome-localized BRAF(C3A) induced seedling lethal and accumulation of ubiquitin conjugates, like FREE1-RNAi line. Based on these results, authors proposed a novel regulatory mechanism of ESCRT function by BRAF; BRAF regulates FREE1 function by competitively binding to Vps23. This model is novel and their results include important findings, but physiological importance of BRAF is still unclear, because knock out mutant of BRAF exhibited WT-like phenotype. In addition, I feel further evidence would be required for the conclusion.

Comments

Authors showed molecular function of endosome-localized BRAF, but did not show the evidence that dysfunction of endosome-localized BRAF, not plasma membrane localized BRAF, induced suppression of FREE1-RNAi. Result of complementation test of *sof524* with endosome-localized BRAF(C3A) need to be added.

According to authors model, overexpression of BRAF(C3A) might affect the localization of FREE1. Data of CM/CS ratio of FREE1 in BRAF(C3A) overexpressor need to be added.

Figure 5A indicate that the number of GFP-FREE1 punctae increased in *braf* mutant. If GFP-FREE1 punctae in *braf* mutant are MVB/PVC and *braf* mutation affect only FREE1 localization, colocalization between FREE1 and MVB/PVC marker should increase in *braf* mutant, but result in Figure 5B did not support this. Please add the data of effect of *braf* mutation to the localization of other organelle markers.

Reviewer #4 (Remarks to the Author):

This manuscript reports on the characterization of a mutant identified through a suppressor screen for the lethal phenotype of a FREE1 knockdown mutant. FREE1 is a plant-specific protein that controls ESCRT machinery for the formation and function of MVBs. The identified mutation maps in a plant-specific protein named BRAF and reverts the phenotype due to the loss of FREE1. The manuscript is potentially interesting the results are novel and, if consolidated, they could enrich knowledge of membrane trafficking regulators in plants for which not much is known.

The paper is generally well written but some additional experiments may help consolidate the author's claims. The main concern is however that BRAF is clearly localized to the PM and much less to the cellular structures where it is postulated to operate. The interaction data in support that BRAF may be an ESCRT component are interesting but need additional controls. Finally, if BRAF competes for FREE1 binding to membranes, its overexpression should give a phenotype similar to a FREE1 loss of function. This is not tested directly and seems to be an important point.

FREE1 is clearly localized to dots in the cell (Fig. 1A) but BRAF is mostly at the PM. Only occasionally it colocalizes with FREE1. This evidence rings the possibility that BRAF may have other functions than ESCRT, which may at least partly explain the observed phenotype.

The interaction of BRAF with VPS23 would benefit from additional controls. While the Y2H analysis is well performed, the other analyses require more appropriate controls. In Figure 1C the input of VPS23 is very high but the co-precipitated BRAF is little. Furthermore, the other assays are done in protoplasts that overexpress the proteins. While this is quite a customary approach, the conclusions on the specific interaction could be reinforced if controls different from cytosolic proteins were provided, as VPS23s and BRAF appear to be associated onto membranes. One also notes the differences in binding with Figure 4C where the BRAF-VPS23A interaction is much stronger than the apparent strength in Figure 3C. Please explain this difference. The data may be improved by introducing a quantification of the strength of interaction of BRAF, BRAF(A330V) and FREE1 for VPS23A.

The authors propose that Vps23 CC domain can serve as an interacting surface for both FREE1 and BRAF binding. They then test competition for the binding of VPS23A BRAF and FREE1 and establish that their binding to VPS23A is mutually exclusive. The argument could be more compelling if they used VPS23A CC in place of the full-length VPS23A for the competition analyses.

The authors tested whether depletion of BRAF in plants increases FREE1 recruitment to endosomes. Figure 5A shows an apparent increase of GFP-FREE1 in the *braf-2* mutant compared to WT cells. However, the *braf-2* images seem to be more cortical than WT. Indeed in WT the more inner cells are visible and those are not visible in *braf-2*. Although the images are 3D reconstructions, the cortical focal plane should be at an identical level for WT and *braf-2*.

The results that FREE1 is recruited more onto the membranes when its expression is induced in the presence of BRAF(A330V) are a bit puzzling. The authors demonstrate in Figure 4 that BRAF(A330V) binding to VAP23A is 5 fold reduced compared to WT BRAF. Hence, although at a limited extent, the mutated protein should be still able to compete with FREE1 for binding to VPS23A. Curiously, what we see is that in *sof524* the levels of FREE1 are reduced compared to WT despite the anti-VSR signal being the same. Is the *sof524* mutation destabilizing FREE1 cellular levels? Could this explain the phenotype?

The claims that the phenotype with the de-acylated BRAF are specifically linked to the hypothesized cause, it should be demonstrated that BRAF is indeed S-acylated. The relevant figures are difficult to interpret as the mutated protein seems to be associated with the PM - is this the case? Also if this mutated version were dominant as supposed the number of ILVs in DEX::BRAFC3A-YFP should be similar to DEX-FREE1 (Figure 1G). Can the authors comment on this? The putative acylation mutant is used in place of WT BRAF to test whether increasing the cellular levels of BRAF causes an effect on the distribution of FREE1. The use of WT BRAF would have been a better tool to use due to the lack of evidence that the C3A mutation affects the BRAF binding to membrane for the lack of acylation.

Minor point:

Page 3, line 22: Please rephrase: "Consistent with the mutant phenotype which malfunction of the assembly or dissociation of the ESCRT machinery",

Reviewer #5 (Remarks to the Author):

The manuscript entitled "A Unique Plant Bro1-Domain Protein, BRAF, Regulates Multivesicular Body Biogenesis and Membrane Protein Homeostasis" reports the characterization of a Bro-1 domain protein, BRAF, identified in a screening that partially reverts the deficient growth and molecular phenotypes of FREE1 RNAi. The mutagenized RNAi line possessed a mutation in the BRAF coding region (A330V) that was complemented by the wild type sequence. The knockdown mutants *braf-2* and *braf-3* did not display a noticeable phenotype (even though stress conditions were not assayed). Nevertheless the double *braf-2xFREE1* RNAi lines showed similar reversion of phenotypes. BRAF-GFP is described to be co-localizing with FM4-64 at the plasma membrane and with the marker mCherry-Rha1 at MVB/PVCs. BRAF-mRFP partially co-localized with GFP-FREE1. By Y2H experiments the authors determined that BRAF and FREE1 they are not directly interacting but both of them interact with the retromer subunits Vps23A and Vps23B. The interactions were confirmed by in vitro binding assays and also IPs. The authors also found that Vps23 coiled-coil domain was important for the interaction with both BRAF and FREE1 and that both proteins compete for this binding site. Interestingly, BRAF A330V showed decreased interaction with Vps23 leading to an increase in the binding of FREE1. This was confirmed using *braf-2xGFP-FREE1* plants where an increase of GFP-FREE1 compartments was detected.

The authors then provide information that is not in complete agreement with their conclusions.

1. On page 15 the text reads, "These results indicate that the increased recruitment of GFP-FREE1 to MVB/PVCs is specifically caused by depletion of BRAF. Such a disturbance in FREE1 membrane distribution can also be detected in the DEX treated *sof524* plants when probed for endogenous FREE1 distribution in cellular fractions. Distinct from WT and FREE1-RNAi plants, the RNAi decreased FREE1 protein in the *sof524* mutant was mainly distributed in the membrane fraction while BRAF (A330V) mutant was mainly in the soluble fraction (Fig.5C)."

BRAF is mainly present at the plasma membrane but also in discrete MBV/PVCs (Fig 3). Additionally, BRAF is present in soluble and membrane protein fractions in WT plants (Fig 5). Following the logic of the authors it would be expected that in the FREE1-RNAi background BRAF would be more strongly associated with MVB/PVCs. However, this seems not to be the case as BRAF remains unchanged in the membrane fraction indicating that lack of FREE1 did not increase association with the membrane fraction. An increase in the association could be masked by the major presence of BRAF in the plasma membrane but this fact is not mentioned in the text. On the contrary, when BRAF is mutated (BRAF A330V) in the FREE1-RNAi background membrane association becomes decreased indicating that the mutation affects the association of BRAF mainly with the plasma membrane. The authors also show an increased presence in the membrane fraction of the remaining FREE1 protein in the *sof524* mutant DEX induced (Fig 5) indicating that a poor association of BRAF with both membranes does not interfere with FREE1 association, on the contrary seems to improve it. There are no images of the subcellular localization of BRAF (A330V) that would greatly help to figure out if the remaining protein observable in the membrane fraction is indeed still present at the MVB/PVC (or not) in the FREE1-RNAi background. The authors fail to address these discussions and propose a unique hypothesis that does not fit all the facts.

2. Regarding the association of BRAF with the plasma membrane the authors produced a version of the protein BRAF (C3A)-YFP that possesses a mutation in the S-acylation site. This mutant is no longer associated with the plasma membrane and is mainly present in the cytosol and in Cerulean-Vps23A or B and mCherry-Rha1 compartments indicating a MVB/PVC localization (Fig 3, 6, Suppl 8).

Interestingly, this protein does not have a MVB/PVC specific localization as does WT BRAF; it co-localizes with the Golgi marker mCherry-SYP32, but not the TGN marker VHAa-1-RFP. This fact that is not mentioned or explained in the text but is included in the supplementary images. Additionally, overexpression of UBQ10pro::BRAF(C3A)-YFP leads to a lethal phenotype. This could indicate both that the association to the plasma membrane is important for function (it is not shown in this case that there is less FREE1 binding) and that the association with Vps23 could be preventing the FREE1 binding generating phenotypes similar to FREE1 mutants like reduced number of ILVs in the

MVB/PVCs. Nevertheless the text indicates on page 8 that "All together, these results support our initial speculation that BRAF affects MVB/PVC formation and the vacuolar degradation of membrane cargo probably by regulating FREE1 recruitment to MVB/PVCs."

The quality of the data is very good, the manuscript is generally well-written and this is an expert lab in plant cell biology. However, the results overall do not bring light to the function of BRAF in the formation of MBV/PVC. Rather it indicates that its absence from MBV/PVC is somehow beneficial for FREE1 association that is indeed implicated in MBV/PVC biogenesis and vacuolar fusion. This would be in agreement with the DEX::BRAF(C3A, A330V)-YFP double mutant that presents a completely cytoplasmic localization and no associated phenotypes. If the authors can demonstrate a direct relationship between BRAF and MBV/PVC biogenesis or even better a mechanistic explanation of the role of BRAF in regulating FREE1 association this will improve the manuscript. Probably stress conditions applied to the different mutants could bring some light in this respect. As is it is intriguing that BRAF could be involved in a novel regulatory mechanism relevant for the plant and general community but this needs to be more clearly demonstrated.

Responses to reviewers:

Reviewer #1 (Remarks to the Author):

Shen et al. describes the unique plant Bro1-domain containing protein BRAF that is responsible for the recruitment of FREE1 to the MVB by competing with FREE1 to bind the CC-domain of the ESCRT-I component VPS23. To my knowledge, these findings are completely new. A novel plant ESCRT-I component is introduced and thus this publication provides essential information for people in plant science but also beyond the plant field. The described competitive binding between proteins of the ESCRT-I machinery sheds light on the possibility of a highly transient ESCRT machinery in plants that will influence scientists working with the ESCRT machinery. The experiments are well planned and the results are highly underlined by a huge amount of negative controls.

Response: Thank you for your positive comments and your time in reviewing our Ms.

The manuscript is clearly written except the legends: There could be more information included. e.g. Suppl. Fig. 1: what is cFBPase and why was it used as reference protein?

Response: We have now included more information about the cFBPase in the Figure legend of Fig. 1C and Suppl. Fig. 1C as: “The cytoplasmic marker anti-cFBPase, a ubiquitously expressed cytosolic fructose-1,6-bisphosphatase, is used as a loading control.” We have also checked and included more detailed information in legends of other Figures in the Ms. (all the changes have been highlighted in red).

Fig. 5, the meaning of CM is described in the method part, but it would be more convenient to have the description in the legend;

Response: We have now added the description of the CM and CS in the Figure legend of Fig. 5C as: "... the loading control of anti-VSR as cell membrane (CM) fraction and anti-cFBPase as cell soluble (CS) fraction, ...".

Suppl. Fig 3; A, it is not clear which light blue bar and what is the meaning of the blue line, and additional bar;

Response: We have re-highlighted the light blue bar in the new Supplementary Fig. 3A and described the meaning of lines and bar in the Figure legend as: "Allele frequency (AF) analysis result of *sof524* on chromosome 5. One specific Col-allele peak appears in the left region as highlighted by the light blue bar, which is predicted as mapping interval. Red circle dots indicate allele frequency estimations on individual markers. The blue line shows the average allele frequency estimations within 200 kb windows with a 5 kb step size. The dashed line in grey indicates window-based boot value of allele frequency (= summation-of-single-marker-AF divided by number-of-markers with minimum quality score involved)."

Suppl. Fig. 5: which tissue? VAMP711 is marking the tonoplast (not indicated here).

Response: Confocal images were taken from the root epidermal cells of 5-d-old Arabidopsis seedlings. We have now included this information in the Figure legend of Supplementary Fig. 5. We have also indicated mCherry-VAMP711 as a tonoplast marker in new Supplementary Fig. 5C.

Methods are clearly described and Shen et al. provides sufficient methodological details. I would suggest following additional experiments to strengthen the paper concerning the localization of BRAF at MVBs and to clearly identify the GFP-FREE1 signal:

.) Shen et al. speculates that BRAF localizes at MVBs by showing a co-localization of BRAF and FREE1 (Fig. 3A). Here, an additional experiment would characterize the co-localization, especially, when the BRAF-mRFP is that weak; for example an inhibitor treatment with Wortmannin would clearly define the structure as MVB.

Response: We have now performed new experiments by treating the plants with Wortmannin (Wort), and the obtained results showed that both GFP-FREE1 and BRAF-mRFP localized to the surface of the enlarged MVBs that appeared as ring-like structures (new Fig. 3B; Page 12, Lines 2 - 7), suggesting that both proteins colocalized on MVB/PVCs.

.) Fig. 5:, A, B: It seems that there is additional GFP-FREE1 signal in huge ring like structures or bubbles which were not mentioned. What are these structures? There is also some lumen structure in the in vivo pictures (A). Are these artefacts? It would strengthen the in vivo study (A) using FM4 to see if the cells are still alive.

Response: The additional GFP-FREE1 signal in “huge ring like structures or bubbles” are nucleus in Arabidopsis root epidermal cells. We have now included a new Supplementary Fig. 8A, showing the nucleus localization of the GFP-FREE1 fluorescence signal with the corresponding DIC image. (See also next response)

According to your suggestion, we have performed new experiments using FM4-64, showing that the cells are still alive (new Fig. 5A).

.) Suppl. Fig. 7: A: Again, the GFP-FREE1 signal is accumulated in big bubbles. Is this the nucleus? Vacuole? This hasn't been shown in recent publications and not in Figure 3A. B: it would strengthen the results if FM4 would be used here (like in A) to see the cell membranes and thus use FM4 as viable marker. Is the scale in B really the same as in A? It seems that the cells are bigger in B than in A.

Response: The additional GFP-FREE1 signal in “big bubbles” are nucleus (please also see the above response). The nucleus localization of GFP-FREE1 signal was also shown in our previously published confocal images (e.g. Fig. S2,A in Gao *et al.*, Curr. Biol., 2014). The nucleus localization signal was not shown in the confocal images of Fig. 3A,B, because these images were collected in more cortical layer.

We have also performed the suggested confocal experiments using FM4-64, showing the cell membranes (new Supplementary Fig. 8B). We have double-checked the scale bars in the original Suppl. Fig. 7A, B, and they are correct. The original Suppl. Fig. 7B has now been replaced by the new Supplementary Fig. 8B with FM4-64 staining.

Minor revisions:

page 3: line 11: ‘with’ should be deleted

Response: Done.

page 5: The last conclusion sentence is formulated in a highly negative way: something is shown that this is not affecting something that negatively affects something. I would prefer a positive message here.

Response: This sentence has been changed as: “Taken together, these results suggested that the reverse phenotype of *sof524* is not caused by a disruption of the RNAi process.” (Page 5, Lines 20 - 22).

page 6: *free1* is described with altered ILV formation and causing accumulation of ubiquitinated membrane cargo in endosomes and at the tonoplast. It is thus supposed, that Shen et al. shows here some microscopic data but Suppl. 2A shows a schema to summarize

this phenotype. This should be somehow rewritten.

Response: We have revised this sentence as: “Ubiquitinated membrane proteins are sorted into the ILVs of MVB/PVCs for further degradation upon MVB/PVC-vacuole fusion. According to our previously proposed model of FREE1 functions in the ILV formation and ubiquitinated membrane cargo sorting (Supplementary Fig. 2A)^{12, 13}, depletion of FREE1 results in failure of ILV formation and consequently causes accumulation of ubiquitinated membrane cargo in endosomes and finally at the tonoplast.” (Page 6, Lines 1 - 6).

page 6: line 13: I would prefer ‘at’ the vacuolar membrane. This would be true for the whole manuscript (e.g. page 11, line 14.)

Response: We have changed into “at the vacuolar membrane” throughout the manuscript.

page 9: line 21/22: What is the S in the BRAFSpro.....?

Response: We have now deleted the “S” and checked throughout the Ms.

page 21: line 2: ‘seedlings’ two times

Response: We have now deleted the first “seedlings”.

page 27: line 8: Yeast two-hybrid not yeast-two hybrid Legends need more detailed description.

Response: We have revised into “yeast two-hybrid” and checked throughout the Ms. We have now provided more detailed description of Y2H analysis in the Figure legends of Fig. 3C, 4B, and Supplementary Fig. 6A.

Thus, I would like to invite the authors to revise their manuscript to address these specific

concerns and I would be very happy to review again the resubmission.

Kind regards,

Verena Ibl

Reviewer #2 (Remarks to the Author):

The manuscript by Shen et al., report a plant unique regulatory mechanisms of MVB/PVC, which is mediated by proteins with Bro1-domain. Authors showed that *sof524* mutant rescues phenotypes of FREE1-RNAi, and responsible gene of *sof524* encodes plant unique type Bro1-domain containing protein, named BRAF. In addition, they showed that BRAF localized on the plasma membrane and FREE1-positive endosomes, and BRAF compete with FREE1 for Vps23 binding on the endosome. Finally, they showed overexpression of endosome-localized BRAF(C3A) induced seedling lethal and accumulation of ubiquitin conjugates, like FREE1-RNAi line. Based on these results, authors proposed a novel regulatory mechanism of ESCRT function by BRAF; BRAF regulates FREE1 function by competitively binding to Vps23. This model is novel and their results include important findings, but physiological importance of BRAF is still unclear, because knock out mutant of BRAF exhibited WT-like phenotype. In addition, I feel further evidence would be required for the conclusion.

Response: Thank you for your comments and your time in reviewing our Ms.

Comments

Authors showed molecular function of endosome-localized BRAF, but did not show the evidence that dysfunction of endosome-localized BRAF, not plasma membrane localized

BRAF, induced suppression of FREE-RNAi. Result of complementation test of *sof524* with endosome-localized BRAF(C3A) need to be added.

Response: Thank you for your suggestion. We have now performed the suggested complementation experiments by transforming *sof524* with endosome-localized *BRAF(C3A)* and cytosol localized *BRAF(C3A,A330V)* as control. The obtained results showed that *BRAF(C3A)/sof524* plants showed seedling lethality phenotype (after DEX treatment) similar to the *FREE1-RNAi*, however, *BRAF(C3A,A330V)/sof524* plants showed a survival phenotype (new Fig. 7D; Page 19, Lines 12 - 22), thus suggesting that the endosome-localized BRAF(C3A) can complement *sof524*, and indicating that the dysfunction of endosome-localized BRAF, rather than the plasma membrane-localized BRAF, induced the suppression of *FREE-RNAi*.

According to authors model, overexpression of BRAF(C3A) might affect the localization of FREE1. Data of CM/CS ratio of FREE1 in BRAF(C3A) overexpressor need to be added.

Response: We have now performed the suggested experiments and analyzed the CM/CS ratio of FREE1 in DEX induced *DEX::BRAF(C3A)-YFP* seedlings. The obtained results showed that less FREE1 was found in the CM fraction in the *DEX::BRAF(C3A)-YFP* plants vs. the WT plants (new Fig. 7A; Page 18, Lines 17 - 22), indicating that overexpression of BRAF(C3A) affect the FREE1 membrane association.

Figure 5A indicate that the number of GFP-FREE1 punctae increased in *braf* mutant. If GFP-FREE1 punctae in *braf* mutant are MVB/PVC and *braf* mutation affect only FREE1 localization, colocalization between FREE1 and MVB/PVC marker should increase in *braf* mutant, but result in Figure 5B did not support this.

Response: Thank you for your suggestion. The reviewer raised a good question about

whether more FREE1 recruitment would increase the colocalization between FREE1 and VSR-positive MVB in *braf* mutant vs. WT. Because “Pearson–Spearman Correlation Coefficient” (used in the original Figure) measures the pixel-by-pixel covariance in the signal levels of two channels, it is not suitable to compare the “colocalization increase” in *braf* vs. WT. To address this, we have performed additional statistical analysis to calculate the “colocalization percentage”, according to previously described methods (Kalinowska et al., PNAS. 2015; Kolb et al., Plant Physiol, 2015). The new calculation data showed that the percentage of colocalization was significantly increased in *braf-2* ($71.4 \pm 5.6\%$, $n=1265$) vs. WT ($58.5 \pm 6.5\%$, $n=1245$). Thus, all these data suggested that the colocalization between FREE1 and MVB/PVC marker are increased in *braf-2* vs. WT. These new data are now included in the new Fig. 5B (Page 15, Lines 14 - 16), whereas the method for calculating the percentage of colocalization has also been included in the Method part (Page 31, Lines 10 - 14).

Please add the data of effect of *braf* mutation to the localization of other organelle markers.

Response: We have performed new experiments to test the possible effects of BRAF mutation on the localization of MVB/PVC marker (YFP-Rha1) and the TGN marker (VHA-a1-GFP). The new data showed that the YFP-Rha1 and the VHA-a1-GFP exhibited identical patterns with similar punctae numbers in WT and *braf-2* mutant, suggesting that the BRAF mutation did not affect the localization of these two organelle markers (new Supplementary Fig. 8C-E; Page 15, Line 20 to Page 16, Line 3).

Reviewer #4 (Remarks to the Author):

This manuscript reports on the characterization of a mutant identified through a suppressor

screen for the lethal phenotype of a FREE1 knockdown mutant. FREE1 is a plant-specific protein that controls ESCRT machinery for the formation and function of MVBs. The identified mutation maps in a plant-specific protein named BRAF and reverts the phenotype due to the loss of FREE1. The manuscript is potentially interesting the results are novel and, if consolidated, they could enrich knowledge of membrane trafficking regulators in plants for which not much is known.

The paper is generally well written but some additional experiments may help consolidate the author's claims.

The main concern is however that BRAF is clearly localized to the PM and much less to the cellular structures where it is postulated to operate. The interaction data in support that BRAF may be an ESCRT component are interesting but need additional controls. Finally, if BRAF competes for FREE1 binding to membranes, its overexpression should give a phenotype similar to a FREE1 loss of function. This is not tested directly and seems to be an important point.

Response: Thank you for your comments and your time in reviewing our Ms. We have now performed all the suggested experiments to address the main concerns (see detailed responses below).

FREE1 is clearly localized to dots in the cell (Fig. 1A) but BRAF is mostly at the PM. Only occasionally it colocalizes with FREE1. This evidence rings the possibility that BRAF may have other functions than ESCRT, which may at least partly explain the observed phenotype.

Response: We agree that BRAF localized to punctae dots and at the PM. However, the punctae dots mainly colocalized with FREE1 on the MVB/PVCs (Fig. 3A,B). In addition, we agree that the PM-localized BRAF may have other functions. Therefore, in the revised Ms, we have included a working model of BRAF function in new Supplemental Fig. 10, with proper

descriptions and a discussion of BRAF function at the MVB/PVCs membrane and its PM localization. (Page 22, Lines 1 - 20).

The interaction of BRAF with VPS23 would benefit from additional controls. While the Y2H analysis is well performed, the other analyses require more appropriate controls. In Figure 3C the input of VPS23 is very high but the co-precipitated BRAF is little. Furthermore, the other assays are done in protoplasts that overexpress the proteins. While this is quite a customary approach, the conclusions on the specific interaction could be reinforced if controls different from cytosolic proteins were provided, as VPS23s and BRAF appear to be onto membrane.

Response: According to your suggestion, we have performed new experiments with ESCRT-I component Vps28A as an additional negative control in pull-down assay and FRET analysis (new Fig. 3D,E; Page 13, Lines 2 - 11). In the co-IP assay, we have also included membrane associated ESCRT proteins (ESCRT-I component Vps28A and Vps37A, ESCRT-III component SNF7A and SNF7B) as negative controls (Supplemental Fig. 6C). The obtained results demonstrated that BRAF specifically interacts with ESCRT-I component Vps23 but not with the controls, thus suggesting the specific interaction between BRAF and Vps23.

One also notes the differences in binding with Figure 4C where the BRAF-VPS23A interaction is much stronger than the apparent strength in Figure 3C. Please explain this difference.

Response: The difference of binding strengths in Figure 4C and original Figure 3C was caused by the different film exposure time in the two sets of experiments. In the revised Ms, the original Fig. 3C has been replaced by the new Fig. 3D with new data in BRAF-Vps23A pull-down using Vps28A as an additional control, according to your suggestion. The new data showed similar binding strengths in new Fig. 3D and Fig. 4C.

The data may be improved by introducing a quantification of the strength of interaction of BRAF, BRAF(A330V) and FREE1 for VPS23A.

Response: We have performed two more independent experiments for the relative quantification analysis of the pull-down efficiency in the competition assay, and have showed the strength of interaction of BRAF, BRAF(A330V) and FREE1 for VPS23A in new Fig. 4D. The obtained data further suggested that (1) BRAF competes with FREE1 in binding to Vps23; and (2) BRAF(A330V) has a lower competition ability to Vps23 than BRAF (Page 14, Lines 13 - 18).

The authors propose that Vps23 CC domain can serve as an interacting surface for both FREE1 and BRAF binding. They then test competition for the binding of VPS23A BRAF and FREE1 and establish that their binding to VPS23A is mutually exclusive. The argument could be more compelling if they used VPS23A CC in place of the full-length VPS23A for the competition analyses.

Response: We have now performed new experiments using Vps23A CC domain VPS23A(CC) for the competition assay. These new data are consistent with the data from using full-length VPS23A, thus suggesting that BRAF competed with FREE1 for ESCRT-I component Vps23 binding through CC domain (new Fig. 4C and new Supplementary Fig. 7A,B; Page 14, Lines 5 - 8; Page 14, Lines 18 - 20).

The authors tested whether depletion of BRAF in plants increases FREE1 recruitment to endosomes. Figure 5A shows an apparent increase of GFP-FREE1 in the braf-2 mutant compared to WT cells. However, the braf-2 images seem to be more cortical than WT. Indeed in WT the more inner cells are visible and those are not visible in braf-2. Although the images

are 3D reconstructions, the cortical focal plane should be at an identical level for WT and *braf-2*.

Response: We have collected new 3D confocal images (with FM4-64 staining as suggested by Reviewer 1) at the identical focal plane from the root epidermal cells of the basal meristem region. The new data showed an increase of GFP-FREE1 punctate dots in *braf-2* compared to WT cells (new Fig. 5A), thus suggesting that the depletion of BRAF in plants increases FREE1 recruitment to endosomes. The methods for image collection and punctate number counting have also been included in the Methods section (Page 31, Lines 14 - 17).

The results that FREE1 is recruited more onto the membranes when its expression is induced in the presence of BRAF(A330V) are a bit puzzling. The authors demonstrate in Figure 4 that BRAF(A330V) binding to VAP23A is 5 fold reduced compared to WT BRAF. Hence, although at a limited extent, the mutated protein should be still able to compete with FREE1 for binding to VPS23A. Curiously, what we see is that in *sof524* the levels of FREE1 are reduced compared to WT despite the anti-VSR signal being the same.

Response: The major concern raised is “that in *sof524* the levels of FREE1 are reduced compared to WT”. Because the *sof524* mutant was in the genetic background of *FREE1-RNAi*, when applied with DEX, the FREE1 protein level in *sof524* was reduced compared to WT.

Is the *sof524* mutation destabilizing FREE1 cellular levels? Could this explain the phenotype?

Response: It could not explain the *sof524* phenotype, because the *sof524* mutation did not affect the FREE1 cellular levels. The immunoblot data showed that the FREE1 protein levels were similar in WT, *FREE1-RNAi*, and *sof524* seedlings when cultured without DEX induction (Fig. 1C, F and Supplementary Fig. 1C). We have now included a working model of BRAF function in new Supplemental Fig. 10 to explain the *sof524* phenotype (Page 22, Lines

1 - 20).

The claims that the phenotype with the de-acylated BRAF are specifically linked to the hypothesized cause, it should be demonstrated that BRAF is indeed S-acylated. The relevant figures are difficult to interpret as the mutated protein seems to be associated with the PM - is this the case?

Response: Thank you for your suggestions. The major concerns raised are:

- (1) “...it should be demonstrated that BRAF is indeed S-acylated.” We have used Biotin switch method in combination with S-acylation inhibitor 2-bromopalmitate (2-BP) treatment for assaying the BRAF protein S-acylation. The obtained data demonstrated that (A) The endogenous BRAF proteins were S-acylated (new Fig. 6A; upper panel); (B) Treatment of the protein with S-acylation inhibitor 2-bromopalmitate (2-BP) inhibited the S-acylation of endogenous BRAF (new Fig. 6A; middle panel; Page 17, Lines 17 - 21). Thus, all the new data suggested that BRAF is indeed S-acylated. The method for protein S-acylation assay have also been included in the Methods section (Page 29, Line 15 to Page 30, Line 4).
- (2) “...mutated protein seems to be associated with the PM...” The mutated protein, BRAF(C3A)-YFP, is not associated with the PM. BRAF(C3A)-YFP signal showed cytosolic pattern in addition to the intracellular punctae (Fig. 6B,D, and Supplementary Fig. 9B-D).

Also if this mutated version were dominant as supposed the number of ILVs in DEX::BRAF(C3A)-YFP should be similar to DEX-FREE1 (Figure 1G). Can the authors

comment on this?

Response: The numbers of ILVs in MVB/PVCs were significantly reduced in both *DEX::BRAF(C3A)-YFP* (Fig. 7B) and *DEX::FREE1-RNAi* (Fig. 1G) compared with their relative controls, indicating that both proteins involved in the MVB/PVCs biogenesis. In addition, because the numbers of ILVs between *DEX::BRAF(C3A)-YFP* (2.70 ± 1.68) and *DEX::FREE1-RNAi* (1.93 ± 1.23) are not much different, it is not sufficient to make a clear distinction.

The putative acylation mutant is used in place of WT BRAF to test whether increasing the cellular levels of BRAF causes an effect on the distribution of FREE1. The use of WT BRAF would have been a better tool to use due to the lack of evidence that the C3A mutation affects the BRAF binding to membrane for the lack of acylation.

Response: Thank you for your suggestion. The major concerns raised are:

(1) “...C3A mutation affects the BRAF binding to membrane for the lack of acylation” We have included new data, showing that BRAF(C3A) was not S-acylated (new Fig. 6A; lower panel). Since BRAF(C3A) localized in cytosol and MVB/PVC, but not PM (Fig. 6B,D), we thus concluded that BRAF(C3A) mutation affects its binding to plasma membrane due to lack of acylation.

(2) “...use of WT BRAF would have been a better tool...” We have now used the wild-type BRAF overexpression plant to test whether increasing the cellular levels of BRAF would cause an effect on the distribution of FREE1. However, the obtained data showed that wild-type BRAF overexpression did not result in increased endosome-localized BRAF (new Supplementary Fig. 5), when compared with the native promoter driven *BRAFPro::BRAF-YFP* plant (new Fig. 5D). Thus, we used the endosome-localized

BRAF(C3A) to study the increased cellular BRAF effect on the distribution of FREE1.

We have now included proper discussions on this point (Page 17, Lines 4 - 15).

Minor point:

Page 3, line 22: Please rephrase: “Consistent with the mutant phenotype which malfunction of the assembly or dissociation of the ESCRT machinery”,

Response: We have revised this sentence as “Consistent with the ESCRT mutants phenotype, in which the assembly or dissociation of the ESCRT machinery is disrupted, ...” (Page 3, Line 22 to Page 4, Line 1)

Reviewer #5 (Remarks to the Author):

The manuscript entitled “A Unique Plant Bro1-Domain Protein, BRAF, Regulates Multivesicular Body Biogenesis and Membrane Protein Homeostasis” reports the characterization of a Bro-1 domain protein, BRAF, identified in a screening that partially reverts the deficient growth and molecular phenotypes of FREE1 RNAi. The mutagenized RNAi line possessed a mutation in the BRAF coding region (A330V) that was complemented by the wild type sequence. The knockdown mutants braf-2 and braf-3 did not display a noticeable phenotype (even though stress conditions were not assayed). Nevertheless the double braf-2xFREE1 RNAi lines showed similar reversion of phenotypes. BRAF-GFP is described to be co-localizing with FM4-64 at the plasma membrane and with the marker mCherry-Rha1 at MVB/PVCs. BRAF-mRFP partially co-localized with GFP-FREE1. By Y2H experiments the authors determined that BRAF and FREE1 they are not directly interacting but both of them interact with the retromer subunits Vps23A and Vps23B. The

interactions were confirmed by in vitro binding assays and also IPs. The authors also found that Vps23 coiled-coil domain was important for the interaction with both BRAF and FREE1 and that both proteins compete for this binding site. Interestingly, BRAF A330V showed decreased interaction with Vps23 leading to an increase in the binding of FREE1. This was confirmed using braf-2xGFP-FREE1 plants where an increase of GFP-FREE1 compartments was detected.

Response: Thank you for your comments and your time in reviewing our Ms.

The authors then provide information that is not in complete agreement with their conclusions.

1. On page 15 the text reads, “These results indicate that the increased recruitment of GFP-FREE1 to MVB/PVCs is specifically caused by depletion of BRAF. Such a disturbance in FREE1 membrane distribution can also be detected in the DEX treated sof524 plants when probed for endogenous FREE1 distribution in cellular fractions. Distinct from WT and FREE1-RNAi plants, the RNAi decreased FREE1 protein in the sof524 mutant was mainly distributed in the membrane fraction while BRAF (A330V) mutant was mainly in the soluble fraction (Fig.5C).”BRAFF is mainly present at the plasma membrane but also in discrete MBV/PVCs (Fig 3). Additionally, BRAF is present in soluble and membrane protein fractions in WT plants (Fig 5). Following the logic of the authors it would be expected that in the FREE1-RNAi background BRAF would be more strongly associated with MVB/PVCs.

However, this seems not to be the case as BRAF remains unchanged in the membrane fraction indicating that lack of FREE1 did not increase association with the membrane fraction. An increase in the association could be masked by the major presence of BRAF in the plasma membrane but this fact is not mentioned in the text.

Response: We agree with the reviewer “that according to the logic of this study, it is expected

that in the *FREE1-RNAi* background BRAF would be more strongly associated with MVB/PVCs. However, BRAF remains almost unchanged in the membrane fraction (Fig. 5C), indicating that lack of FREE1 did not significantly increase the association with the membrane fraction. An increase in the association could be masked by the major presence of BRAF in the plasma membrane”. Now we have mentioned this point in the main text (Page 16, Lines 11 - 15).

On the contrary, when BRAF is mutated (BRAF A330V) in the FREE1-RNAi background membrane association becomes decreased indicating that the mutation affects the association of BRAF mainly with the plasma membrane.

The authors also show an increased presence in the membrane fraction of the remaining FREE1 protein in the sof524 mutant DEX induced (Fig 5) indicating that a poor association of BRAF with both membranes does not interfere with FREE1 association, on the contrary seems to improve it.

There are no images of the subcellular localization of BRAF (A330V) that would greatly help to figure out if the remaining protein observable in the membrane fraction is indeed still present at the MVB/PVC (or not) in the FREE1-RNAi background. The authors fail to address these discussions and propose a unique hypothesis that does not fit all the facts.

Response: We are grateful to the reviewer for providing these insightful comments. The major concern raised is “no images of the subcellular localization of BRAF (A330V)” in the *FREE1-RNAi*. We have now performed new experiments and studied the subcellular localization of BRAF (A330V) in the *FREE1-RNAi* background. The obtained confocal data showed that the BRAF(A330V)-YFP was mainly localized at PM while the BRAF-YFP was localized at both PM and intracellular punctae (new Fig. 5D; Page 16, Lines 15 - 18), suggesting that the point mutation in BRAF(A330V) affects the association of BRAF mainly

with MVB, but not the PM.

2. Regarding the association of BRAF with the plasma membrane the authors produced a version of the protein BRAF (C3A)-YFP that possesses a mutation in the S-acylation site. This mutant is no longer associated with the plasma membrane and is mainly present in the cytosol and in Cerulean-Vps23A or B and mCherry-Rha1 compartments indicating a MVB/PVC localization (Fig 3, 6, Suppl 8). Interestingly, this protein does not have a MVB/PVC specific localization as does WT BRAF; it co-localizes with the Golgi marker mCherry-SYP32, but not the TGN marker VHAA-1-RFP. This fact that is not mentioned or explained in the text but is included in the supplementary images.

Response: Thank you for your suggestions. We have now described the BRAF localization with the Golgi and TGN markers (Page 18, Lines 13 - 16) with proper discussion in the revised Ms (Page 22, Lines 18 - 20).

Additionally, overexpression of UBQ10pro::BRAF(C3A)-YFP leads to a lethal phenotype. This could indicate both that the association to the plasma membrane is important for function (it is not shown in this case that there is less FREE1 binding) and that the association with Vps23 could be preventing the FREE1 binding generating phenotypes similar to FREE1 mutants like reduced number of ILVs in the MVB/PVCs. Nevertheless the text indicates on page 8 that “All together, these results support our initial speculation that BRAF affects MVB/PVC formation and the vacuolar degradation of membrane cargo probably by regulating FREE1 recruitment to MVB/PVCs.”

Response: We are thankful to the reviewer for the comments. The major concerns raised are:

(1) Whether “the association to the plasma membrane is important for function” in terms of lethal phenotype in BRAF(C3A)-YFP overexpression plants. We do not think so because:

The plant overexpressing BRAF(C3A,A330V)-YFP is survival, in which the protein is also dis-associated from the PM but without MVB/PVCs localization (Fig. 6C,D), thus suggesting that the lethal phenotype of BRAF(C3A)-YFP is not caused by its dis-association from PM.

(2) Whether less FREE1 binding to membrane. We have now included new data showing that less FREE1 was found in the CM fraction in the *DEX::BRAF(C3A)-YFP* plants vs. the WT plants (new Fig. 7A; Page 18, Lines 17 - 22). The new data indicated that *BRAF(C3A)-YFP* prevented the FREE1 binding to membrane, thus resulting in phenotypes similar to *FREE1-RNAi* plants.

The quality of the data is very good, the manuscript is generally well-written and this is an expert lab in plant cell biology. However, the results overall do not bring light to the function of BRAF in the formation of MBV/PVC. Rather it indicates that its absence from MBV/PVC is somehow beneficial for FREE1 association that is indeed implicated in MBV/PVC biogenesis and vacuolar fusion. This would be in agreement with the *DEX::BRAF(C3A, A330V)-YFP* double mutant that presents a completely cytoplasmic localization and no associated phenotypes. If the authors can demonstrate a direct relationship between BRAF and MBV/PVC biogenesis or even better a mechanistic explanation of the role of BRAF in regulating FREE1 association this will improve the manuscript. Probably stress conditions applied to the different mutants could bring some light in this respect. As is it is intriguing that BRAF could be involved in a novel regulatory mechanism relevant for the plant and general community but this needs to be more clearly demonstrated.

Response: Thank you for your comments.

We agree that the stress conditions could be applied to further study other functions of BRAF, and have now included a discussion on this point (Page 22, Line 20 to Page 23, Line 3).

The novel finding of this study is that BRAF **negatively regulates** the formation of ILVs in MVBs and vacuolar degradation of membrane cargo in plants. As a negative ESCRT regulator of FREE1, the absence of BRAF from MBV/PVC is beneficial for FREE1 association while over-accumulation of BRAF to MBV/PVC disturbs FREE1 association. Such regulatory mechanism has not been reported before in other eukaryotes, and it is distinct from the known positive regulators. In addition, we have also included a working model of BRAF function in new Supplemental Fig. 10 with proper description and a discussion on BRAF function at the MVB/PVCs membrane and the nature of its PM localization (Page 22, Lines 1 - 20).

REVIEWERS' COMMENTS:

Reviewer #1 (Remarks to the Author):

The points that I raised were answered in a satisfactory way by including additional experiments. I just recommend the author to read through the manuscript carefully as there are still some spelling arrows (e.g. Y2H versus YTH, VPS versus vps...). Additionally, the western blot in Figure 5C appears a little bit messy as there is some additional signal at the indicated lines of the marker and in the back of the singals in the lane of anti-cFBPase.

Thus, I would like to invite the authors to revise their manuscript to address these specific minor concerns. After revision, I endorse the publication of this manuscript.

Reviewer #2 (Remarks to the Author):

I think that authors addressed to the comments adequately, except for one comment from reviewer 4.

(2) "...use of WT BRAF would have been a better tool..." We have now used the wild-type BRAF overexpression plant to test whether increasing the cellular levels of BRAF would cause an effect on the distribution of FREE1. However, the obtained data showed that wild-type BRAF overexpression did not result in increased endosome-localized BRAF (new Supplementary Fig. 5), when compared with the native promoter driven BRAFPro::BRAF-YFP plant (new Fig. 5D).

Comment to the response:

Supplementary Figure 5 indicates that UBQpro:BRAF-GFP localized at the plasma membrane and intracellular punctae, but amount of BRAF-GFP on punctate compartment is not presented. Thus, it is difficult to compare the localization pattern of UBQPro:BRAF-GFP (Supplementary Fig. 5) and BRAFPro:BRAF-YFP (Fig. 5D). The punctae per field data of UBQPro:BRAF-GFP, like Fig. 5D, might help to compare these figures.

Reviewer #5 (Remarks to the Author):

Shen et al

This is a review of a revised manuscript reporting on the protein BRAF which is involved in MVB regulation along with FREE1 in Arabidopsis.

This is a significant report that provides an avenue toward understanding the mechanisms in the regulation of MVB to vacuole transport. I have no major concerns as the authors have done a thoughtful job of addressing all of my concerns. I appreciate the working model Suppl Fig 10. This helps to summarize to results for the reader. In addition, the demonstration that BRAF is acylated addresses a major reviewer concern.

As a minor comment, the text of the revision could use some editing as there are errors in syntax that could be easily addressed prior to a future submission or re-submission. This will help reviewers in the future.

Responses to reviewers:

Reviewer #1 (Remarks to the Author):

The points that I raised were answered in a satisfactory way by including additional experiments.

Response: Thank you for your positive comments and your time in reviewing our Ms.

I just recommend the author to read through the manuscript carefully as there are still some spelling errors (e.g. Y2H versus YTH, VPS versus vps...).

Response: We have revised “YTH” into “Y2H” and “VPS” into “Vps”, and we have carefully checked the spelling throughout the Ms again (Please see the track changes).

Additionally, the western blot in Figure 5C appears a little bit messy as there is some additional signal at the indicated lines of the marker and in the back of the signals in the lane of anti-cFBPase.

Response: In the revised Ms, the immunoblot image using anti-cFBPase in the original Fig. 5C has been replaced by the new image using the same batch of proteins (new Fig. 5C; anti-cFBPase).

Thus, I would like to invite the authors to revise their manuscript to address these specific minor concerns. After revision, I endorse the publication of this manuscript.

Reviewer #2 (Remarks to the Author):

I think that authors addressed to the comments adequately, except for one comment from reviewer 4.

(2) “...use of WT BRAF would have been a better tool...” We have now used the wild-type BRAF overexpression plant to test whether increasing the cellular levels of BRAF would cause an effect on the distribution of FREE1. However, the obtained data showed that wild-type BRAF overexpression did not result in increased endosome-localized BRAF (new Supplementary Fig. 5), when compared with the native promoter driven BRAFPro::BRAFYFP plant (new Fig. 5D).

Comment to the response:

Supplementary Figure 5 indicates that UBQpro:BRAF-GFP localized at the plasma membrane and intracellular punctae, but amount of BRAF-GFP on punctate compartment is not presented. Thus, it is difficult to compare the localization pattern of UBQPro:BRAF-GFP (Supplementary Fig. 5) and BRAFPro:BRAF-YFP (Fig. 5D). The punctae per field data of UBQPro:BRAF-GFP, like Fig. 5D, might help to compare these figures.

Response: Thank you for the suggestion. We have now quantified the punctae number of BRAF-GFP in the *UBQPro::BRAF-GFP* lines. The new data showed that the punctae numbers in *UBQPro::BRAF-GFP* and *BRAFPro::BRAF-YFP* were similar ($P>0.05$), suggesting that the wild-type BRAF overexpression did not result in increased endosome-localized BRAF. The new data have now been included in the revised MS as new Supplementary Fig. 5G (Page 17, Lines 355-358).

Reviewer #5 (Remarks to the Author):

Shen et al

This is a review of a revised manuscript reporting on the protein BRAF which is involved in MVB regulation along with FREE1 in Arabidopsis.

This is a significant report that provides an avenue toward understanding the mechanisms in the regulation of MVB to vacuole transport. I have no major concerns as the authors have done a thoughtful job of addressing all of my concerns. I appreciate the working model Suppl Fig 10. This helps to summarize to results for the reader. In addition, the demonstration that BRAF is acylated addresses a major reviewer concern.

Response: Thank you for your positive comments and your time in reviewing our Ms.

As a minor comment, the text of the revision could use some editing as there are errors in syntax that could be easily addressed prior to a future submission or re-submission. This will help reviewers in the future.

Response: The Ms has been edited by a native English scientist.